



# Climatology and Surface Impacts of Atmospheric Rivers on West Antarctica

Michelle L. Maclennan[1], Jan T. M. Lenaerts[1], Christine A. Shields[2], Andrew O. Hoffman[3], Nander Wever[1], Megan Thompson-Munson[1], Andrew C. Winters[1], Erin C. Pettit[4], Theodore A. Scambos[5], and Jonathan D. Wille[6]

[1]Department of Atmospheric and Oceanic Sciences, University of Colorado Boulder, Boulder, CO, USA
[2]National Center for Atmospheric Research, Boulder, CO, USA
[3]Department of Earth and Space Sciences, University of Washington, Seattle, WA, USA
[4]College of Earth, Ocean, and Atmospheric Sciences, Oregon State University, Corvallis, OR, USA
[5]Earth Science and Observation Center, Cooperative Institute for Research in Environmental Sciences, University of Colorado Boulder, Boulder, CO, USA
[6]Institut des Géosciences de l'Environnement, Université Grenoble-Alpes, Grenoble, France

**Correspondence:** Michelle L. Maclennan (michelle.maclennan@colorado.edu)

**Abstract.** Atmospheric rivers (ARs) transport large amounts of moisture from the mid– to high–latitudes and they are a primary driver of the most extreme snowfall events on Antarctica. ARs also raise surface temperatures when they make landfall over Antarctica, leading to surface melting. In this study, we characterize the climatology and surface impacts of ARs on West Antarctica, focusing on the Amundsen Sea Embayment and Marie Byrd Land. First, we develop a climatology of ARs in this

region, using an Antarctic-specific AR detection tool combined with MERRA-2 and ERA5 atmospheric reanalyses. We find that while ARs are infrequent, they cause intense precipitation in short periods of time and account for 11% of the annual surface accumulation. They are driven by the coupling of a blocking high over the Antarctic Peninsula with a low–pressure system known as the Amundsen Sea Low. Next, we use observations from automatic weather stations on Thwaites Eastern Ice Shelf to examine a case study of 3 ARs that made landfall in rapid succession from February 2 to 8, known as an AR family

event. We use snow height observations to force the firn model SNOWPACK to reconstruct accumulation and surface melting during the event, and compare these results with accumulation higher up on the glacier derived from surface height changes using interferometric reflectometry. While accumulation dominates the surface impacts of the event on Thwaites Eastern Ice Shelf (>100 kg m$^{-2}$), we find small amounts of surface melt as well (<5 kg m$^{-2}$). West Antarctica currently experiences minimal surface melting, most of which is absorbed by the firn, but future atmospheric warming could lead to more widespread

surface melting in West Antarctica. Combined with a future increase in AR intensity or frequency, this could limit the ability of the firn layer to absorb melt water, which could harm ice shelf stability, and ultimately accelerate mass loss of the West Antarctic Ice Sheet. The results presented here enable us to quantify the past impacts of ARs on West Antarctic surface mass balance and characterize their interannual variability and trends, enabling a better assessment of future AR-driven changes in the surface mass balance.





## 1 Introduction

The Antarctic Ice Sheet (AIS) represents a vast, dynamic system, whose mass is gained over time through snow accumulation and lost through ablation. The balance of these processes, known as the mass balance, is calculated as the difference between mass changes at the surface of the ice sheet, known as the surface mass balance (SMB), and discharge of ice across the grounding line. In the last four decades, the AIS has experienced increased mass loss, from $40 \pm 9$ Gigatons per year (Gt yr$^{-1}$)

between 1979 and 1990 to $252 \pm 26$ Gt yr$^{-1}$ between 2009 and 2017, due to increasing discharge across the grounding line of the West Antarctic Ice Sheet (WAIS Rignot et al., 2019). From 1979 to 2017, the WAIS contributed $6.9 \pm 0.6$ mm of global mean sea level rise (Rignot et al., 2019).

Although it covers only 17% of the AIS, the WAIS accounts for 34% of ice discharge. Within the WAIS, the Amundsen Sea sector has experienced more than a doubling (130% increase) of ice discharge from 1979 to 2017 (Rignot et al., 2019).

In this region, the enhanced flow of relatively warm circumpolar deep water beneath ice shelves on coastal West Antarctica (including Thwaites Eastern Ice Shelf) has increased basal melt of the ice shelves and led to widespread grounding line retreat (Jacobs and Hellmer, 1996; Thoma et al., 2008; Jacobs et al., 2011; Dutrieux et al., 2014; Rignot et al., 2014; Alley et al., 2021; Milillo et al., 2022; Wild et al., 2022). In particular, TG is at considerable risk for continued grounding line retreat in the future because it is grounded on inward sloping bedrock, which may lead to a rapid positive feedback for increasing ice flow

and retreat, termed 'marine ice sheet instability' (Weertman, 1974; Schoof, 2012). This process may be underway already, but a rapid acceleration in the 22nd century could result in several tens of cm of sea level rise (Joughin et al., 2014; DeConto and Pollard, 2016; Seroussi et al., 2017; Shepherd et al., 2018; DeConto et al., 2021).

While discharge from the WAIS has increased since 1979, the SMB has experienced no significant trend in the past four decades, despite its large interannual variability (Medley et al., 2014; Rignot et al., 2019). The SMB represents the balance

between mass gained at the surface through precipitation, and mass lost by sublimation and surface meltwater runoff (Lenaerts et al., 2019). Snowfall acts as the primary contributor to the SMB. At present, rainfall is an insignificant component of the SMB as rainfall over the WAIS is very low, and most refreezes in the firn (Vignon et al., 2021). However, rainfall can lower the surface albedo and thereby increase future melting (Wille et al., 2019). While WAIS SMB has been relatively constant long-term, interannual variability in SMB is high: in the Amundsen Sea sector, SMB variability has the same order of magnitude as

the annual mass loss (Medley et al., 2014; Lenaerts et al., 2018; Donat-Magnin et al., 2020). The SMB in this region is driven by extreme snowfall events, which contribute more than 50% of the total annual snowfall, and help to drive the high interannual variability (Maclennan and Lenaerts, 2021). The combination of high interannual variability and seasonal variability in snowfall drives large variations in the annual SMB of the WAIS. To constrain past and future changes in the mass balance of the WAIS, it is essential to diagnose the character and impacts of these extreme events on the SMB.

Among the primary drivers of these extreme snowfall events are atmospheric rivers (ARs), which only make landfall about 3 days per year in Antarctic coastal regions, but contribute 10–20% of the total annual snowfall to the AIS each year (Wille et al., 2021; Gorodetskaya et al., 2014). ARs are long, narrow bands of warm and moist air that propagate poleward from the extra-tropics and are often associated with extratropical cyclones (Zhu and Newell, 1998). When a cold front catches up with





a warm front in the extratropical cyclone, it sweeps up water vapor in the warm region, forming a narrow band of moist air

ahead of the cold front (Bao et al., 2006; Dacre et al., 2015). ARs are associated with a low-level jet and moisture fluxes on the order of the Amazon River (Zhu and Newell, 1998). When ARs encounter land and are lifted orographically, these bands of warm and moist air produce intense precipitation, which can lead to flooding in coastal regions of the midlatitudes, including California, Western Europe, and the Andes in South America (Zhu and Newell, 1998; Ralph et al., 2006; Viale and Nuñez, 2011; Lavers and Villarini, 2013; Ramos et al., 2015; Waliser and Guan, 2017; Lamjiri et al., 2017; Whan et al., 2020). ARs

account for over 90% of poleward water vapor transport in the mid- and high-latitudes (Nash et al., 2018). The primary driver for AR transport towards the Antarctic continent is atmospheric blocking by high–pressure ridges, which enhance the transport of heat and moisture from low to high latitudes (Terpstra et al., 2021; Pohl et al., 2021).

Antarctic ARs are unique in that they can have multiple, contrasting impacts on the SMB. While ARs make landfall up to 14% of the time in the mid-latitudes (∼50 days per year), they are comparatively rare over the AIS, making landfall only 1%

of the time (or ∼3 days per year, Rutz et al., 2019; Wille et al., 2021). Although they occur infrequently, ARs cause intense precipitation when they make landfall over the AIS, because they carry so much moisture. ARs are associated with strong, localized accumulation events in East Antarctica that can account for up to 80% of the annual SMB (Gorodetskaya et al., 2014). Furthermore, they explain 63% of satellite altimetry-identified increases in surface height in West Antarctica in 2019– 2020 (Adusumilli et al., 2021). The ability of ARs to transport heat poleward is important for the AIS as well, as ARs raise

surface temperatures on the ice sheet through warm air advection and enhanced cloud radiative forcing (downwelling longwave radiation) (Wille et al., 2019). This causes surface melting in coastal Antarctica, particularly on the Antarctic Peninsula, which can lead to runoff and/or deplete the ability of the firn to store future meltwater (Wille et al., 2019; Neff et al., 2014). Unlike the Greenland Ice Sheet (Mattingly et al., 2018), ARs act to increase Antarctic SMB, as they cause significantly more snowfall than surface melting.

Despite the importance of ARs for Antarctic SMB, they are relatively poorly understood phenomena. There is high spatial variability in AR landfall over the AIS, and their impacts vary greatly depending on their duration (Wille et al., 2021). Because ARs occur over short periods of time and have multiple impacts on the SMB, it is useful to examine them from both a large-scale, climatological perspective and a focused, case study perspective. We rely on reanalysis to quantify the total landfalls and impacts of ARs from 1980 to 2020 (Wille et al., 2021), but this limits our understanding of ARs to the spatiotemporal

resolution of reanalysis products. To supplement this analysis, we use in–situ automatic weather station data captured by the Automated Meteorology–Ice–Geophysics Observation System (AMIGOS, Scambos et al., 2013) to provide key insights on in-situ conditions during AR landfall. In this study, we first determine the climatology of ARs that make landfall over the WAIS, focusing on the Amundsen Sea Embayment and Marie Byrd Land, and examine their interannual and seasonal variability. We then examine a case study of a series of three successive ARs that made landfall on TG in February 2020, using reanalysis,

in–situ observations, and a firn model. Finally, we discuss the results in the context of how ARs contribute to the present mass balance of the AIS and how their frequency and precipitation may change in future climate scenarios.





## 2    Data and Methods

### 2.1    Observations from Automatic Weather Stations (AMIGOS)

Through the Thwaites-Amundsen Regional Survey and Network Integrating Atmosphere-Ice-Ocean Processes (TARSAN)
project of the International Thwaites Glacier Collaboration, AMIGOS were installed on Thwaites Eastern Ice Shelf at Cavity
Camp (75.033 °S,105.617 °W) and Channel Camp (75.050 °S,105.4334 °W) during a field campaign in austral summer
2019/20 (Fig. 1). Cavity Camp is located on a flat part of Thwaites Eastern Ice Shelf, whereas Channel Camp sits within the
surface expression of a melt channel. The AMIGOS at Cavity Camp and Channel Camp provide unique in-situ observations
of the weather occurring over Thwaites Eastern Ice Shelf by logging hourly temperature, surface pressure, wind speeds and
direction, humidity, and snow height, as well as other observations such as GPS position and firn temperature. We utilize these
novel observations to characterize the in–situ atmospheric conditions — air temperature, surface pressure, wind speed, and
wind direction — for the case study of an AR that made landfall over TG on February 2, 2020, and compare the independent
AMIGOS observations with the Modern-Era Retrospective Analysis for Research and Applications, version 2 (MERRA-2,
Gelaro et al., 2017) and the European Centre for Medium-Range Weather Forecasts (ECMWF) Reanalysis v5 (ERA5, Hersbach
et al., 2020) reanalyses. The AMIGOS temperature sensor is located about 6 m above the surface during the period of interest in
this study, so we refer to AMIGOS air temperatures as "near-surface" when compared to 2 m air temperatures from MERRA-2
and ERA5. The choice not to correct observed temperatures to 2 m above the surface is justified because the atmospheric
surface layer is usually neutrally stable and well-mixed during AR events. We also use the observations to force the firn
model SNOWPACK (Lehning et al., 2002b, a) to reconstruct accumulation during the case study event. Furthermore, we use
a firn-temperature record from thermistors placed 1 m below the surface at the AMIGOS stations to identify changes in firn
temperature associated with percolation of surface meltwater. Sustained hurricane-force winds of 55–70 m s$^{-1}$ over Thwaites
Eastern Ice Shelf in late September 2021 severely damaged the AMIGOS and have limited the available weather data to a
period of just over 1.5 yrs after installment.

### 2.2    Reanalysis Products: MERRA-2 and ERA5

To characterize the large-scale atmospheric circulation and weather conditions associated with Antarctic AR events, we use
the global atmospheric reanalyses MERRA-2 and ERA5. Reanalyses assimilate observations (not including the AMIGOS
data) with atmospheric modeling to produce regularly–gridded, temporally continuous weather and climate data. Due to the
spatially and temporally limited observations available for the Southern Ocean and Antarctica, reanalysis products are essential
for examining spatial patterns in atmospheric pressure, temperature, and moisture, and the resulting snowfall over the WAIS.
MERRA-2 is run at a latitude by longitude resolution of 0.5° x 0.625° (Gelaro et al., 2017), which is approximately 56 x 18
km at the location of TG. ERA5 is the newest global atmospheric reanalysis product from the European Centre for Medium-
range Weather Forecasts (ECMWF), and is run at a resolution of 0.25° latitude (28 km) by 0.25° longitude (7 km) resolution
(Hersbach et al., 2020).





We use MERRA-2 reanalysis to generate surface pressure and surface pressure anomaly (relative to 1980 to 2020 climatol-
ogy) composite maps during the times of AR landfalls over coastal West Antarctica, including the Amundsen Sea Embayment
and Marie Byrd Land (1980 to 2020). We further use MERRA-2 to examine spatial patterns in surface pressure, 500 hPa
geopotential height, and the 500 hPa geopotential anomaly (relative to 1980 to 2020 climatology) over West Antarctica and the
Southern Ocean during the case study AR event in February 2020. We focus on this region as storm events over the Amundsen
Sea Embayment and Marie Byrd Land are susceptible to meridional synoptic flow, whereas the Antarctic Peninsula is more
often subject to zonal flow (Turner et al., 2013; Gonzalez et al., 2018; Maclennan and Lenaerts, 2021). Furthermore, this region
(which includes TG) has experienced the largest acceleration in mass loss in recent decades (Rignot et al., 2019), and we aim
to quantify the long– and short–term impacts of ARs on the SMB.

Over Thwaites Eastern Ice Shelf, we compare MERRA-2 2 m temperatures before and during AR events, to examine the
impacts of ARs on surface temperatures and melting. We also use MERRA-2 for maps of surface pressure during the AR case
study in February 2020. We compare MERRA-2 and ERA5 to the atmospheric conditions observed by the AMIGOS (near-
surface temperature, surface pressure, wind speed, and wind direction) during the case study event. We also compare snow
accumulation reconstructed from snow height observations to accumulation in MERRA-2 and ERA5 during the event.

### 2.3   Atmospheric River Detection Catalogue and Attributing Precipitation

We use a polar-specific AR detection algorithm produced by Wille et al. (2021) to identify the occurrence and landfall of
ARs over the AIS. The algorithm uses the MERRA-2 atmospheric reanalysis (Gelaro et al., 2017) from 37.5° S to 80° S to
detect moisture filaments when integrated water vapor (IWV) or meridional integrated vapor transport (vIVT) exceed the 98th
percentile of their monthly climatologies. Moisture filaments are classified as ARs if they extend at least 20° in the meridional
direction. Because it tracks IWV and vIVT relative to the monthly climatology, this algorithm is uniquely positioned to detect
polar ARs. Wille et al. (2021) found that IWV is better suited for identifying ARs that cause surface melting, as high IWV
over the AIS is associated with cloud development and high downwelling longwave radiation to the surface. Comparatively,
the vIVT-based definition of ARs is better suited for studying snowfall, since the meridional transport of water vapor is linked
to atmospheric dynamics that lead to precipitation.

In this study, we use the vIVT catalogues, with AR detection at 3 hourly intervals using MERRA-2 reanalysis. From the
catalogues, we generate maps of AR frequency over the AIS. We then identify individual AR events making landfall over the
Amundsen Sea Embayment and Marie Byrd Land by counting the number of consecutive time indices of AR landfall within
the region. If the time steps indicate a 12 hour break or more, we count them as two separate AR events. We do this to account
for AR families, which are multiple ARs that make landfall in succession in short periods of time (Fish et al., 2022). Based on
the AR events we identified, we quantify their trends, duration, and seasonality from 1980 to 2020. To attribute precipitation to
ARs, we integrate MERRA-2 3 hourly precipitation within the footprint of the AR landfall, based on the vIVT AR catalogue
of Wille et al. (2021). We also attribute precipitation that falls up to 24 hours after landfall, but within the footprint of the AR,
to the AR event.





## 2.4 SNOWPACK Firn Modeling

We determine the accumulation attributed to the case study AR event of February 2020 using two distinct approaches. In the first, we use precipitation from MERRA-2 and ERA5 reanalyses. In the second, we use observed snow height from the AMIGOS to force the firn model SNOWPACK to reconstruct accumulation during the AR event. SNOWPACK is a physics-based, multi-layer firn model (Lehning et al., 2002b, a), which has been extensively applied in polar regions (Groot Zwaaftink et al., 2013; Steger et al., 2017; Van Wessem et al., 2021; Keenan et al., 2021). The model calculates fresh accumulation density as a function of meteorological conditions, particularly wind and the presence of drifting snow (Wever et al.). We configure the SNOWPACK model here to interpret increases in observed snow height as snowfall, when relative humidity, air temperature and snow surface temperature meet snowfall conditions (Lehning et al., 1999; Wever et al., 2015). This is then combined with fresh accumulation density, to convert to accumulated mass.

We use the MeteoIO library (Bavay and Egger, 2014) to preprocess the meteorological forcing data. Since the measured snow depth exhibited oscillations, seemingly synchronized with the diurnal cycle in temperature, we applied a weighted moving average smoothing filter with a centered window of 48 hrs. Both the Cavity Camp and Channel Camp AMIGOS are represented by the same closest MERRA-2 grid point (Fig. 1). However, we run SNOWPACK individually for each of the AMIGOS. Using the model settings as described in (Keenan et al., 2021), we run SNOWPACK for the period 1980 to the installation date of the respective AMIGOS. We then mark the layer in the SNOWPACK simulated firn, whose depth corresponds to the measured position of the base plate of the AMIGOS upon installation in the field. In the remainder of the simulation, we track the position of this marked layer, and the AMIGOS snow depth is referenced to this specific layer. We can then use the aforementioned snow height driven accumulation approach, while eliminating the settling of the AMIGOS station in the firn column. From the installation date of the AMIGOS forward, we use the available parameters from the AMIGOS to force the simulation (i.e., snow depth, near-surface air temperature, relative humidity and wind speed), while taking downwelling shortwave and longwave radiation from MERRA-2.

## 2.5 Surface Height Changes via Interferometric Reflectometry

We supplement the record of surface height change estimates observed by the AMIGOS on Thwaites Eastern Ice Shelf with surface height change measurements observed with the global navigation satellite system (GNSS) using interferometric reflectometry (Larson et al., 2009, 2015; Roesler and Larson, 2018). TG has two GNSS receiver sites located 124 km (GNSS Lower Thwaites) and 215 km (GNSS Upper Thwaites) from the present grounding zone (Fig. 1, Wilson et al., 2009a, b). GNSS Lower Thwaites is at an elevation of 1011 m and GNSS Upper Thwaites is at an elevation of 1315 m. Using the signal-to-noise ratio of multi-path interference with the direct signal, we determine the geometry of the reflection surface and map the evolution of the snow surface with time. Forward models of reflector height change due to snow loading and densification combined with inverse methods capable of sampling posterior distributions of model parameters using likelihood estimates that assimilate reflector height data can be used to solve for the joint accumulation and densification. From these independent records of accumulation and deposition, we can verify the influence of atmospheric river landfall on observed surface mass balance. We



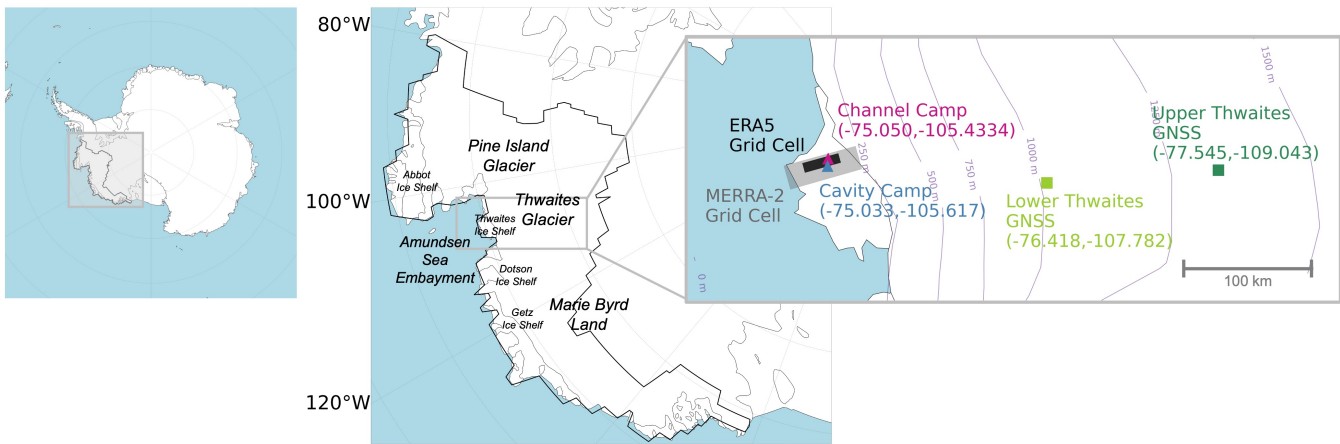

**Figure 1.** Map showing the region of interest in this study – the Amundsen Sea Embayment and Marie Byrd Land in West Antarctica. Locations of the AMIGOS at Cavity Camp (blue) and Channel Camp (pink) are shown on Thwaites Eastern Ice Shelf, along with the MERRA-2 (grey) and ERA5 (black) grid cells nearest to the AMIGOS. Also shown are the GNSS receiver sites at Lower Thwaites (light green) and Upper Thwaites (dark green), located on the grounded Thwaites Glacier. In this figure, and any following maps, elevation contours are from MERRA-2 (doi:10.5067/ME5QX6Q5IGGU) and drainage basins are from Zwally et al. (2012).

show GNSS-derived accumulation with the SNOWPACK-reconstructed accumulation to examine the inland propagation of the case study AR event from lower TG (AMIGOS and Lower Thwaites GNSS) to upper TG (Upper Thwaites GNSS). These observations provide additional spatial information on accumulation patterns due to the AR event.

## 3 Results

### 3.1 Climatology of West Antarctic Atmospheric Rivers

Our analyses show that ARs exhibit hot spots in activity over the Amundsen Sea Embayment and Marie Byrd Land, with a frequency of 3.2% of the total time from 1980 to 2020 (Fig. 2). This represents the total frequency of ARs over the region, calculated by dividing the number of AR times by the total time from 1980 to 2020. Within the region, localized AR frequencies range from 0.2 to 0.8% of the time, with the highest frequencies over the Abbot Ice Shelf and the Getz Ice Shelf. Integrated over the entire region, ARs contribute $59 \pm 24$ Gt precipitation annually (out of $550 \pm 63$ Gt total annual precipitation), and explain 28.7% of the interannual variability in precipitation (linear trends removed). The correlation between AR precipitation and the total annual precipitation is moderately positive, with Pearson correlation coefficient $r = 0.52$ $(p < 0.01)$ (Supplementary Fig. A1).



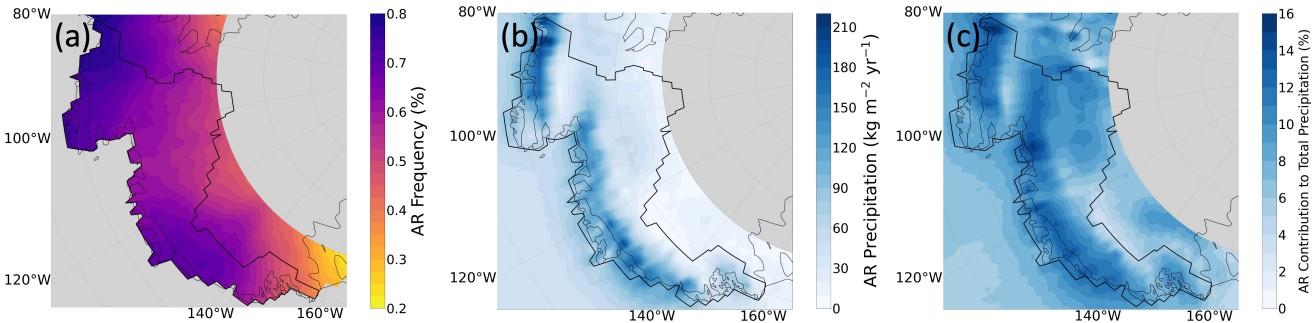

**Figure 2.** (a) Temporal frequencies of AR landfall, (b) precipitation attributed to ARs, and (c) AR precipitation as the percentage of the total annual precipitation in West Antarctica (MERRA-2, 1980 to 2020). The Amundsen Sea Embayment and Marie Byrd Land (region of interest) are outlined in black in all three figures.

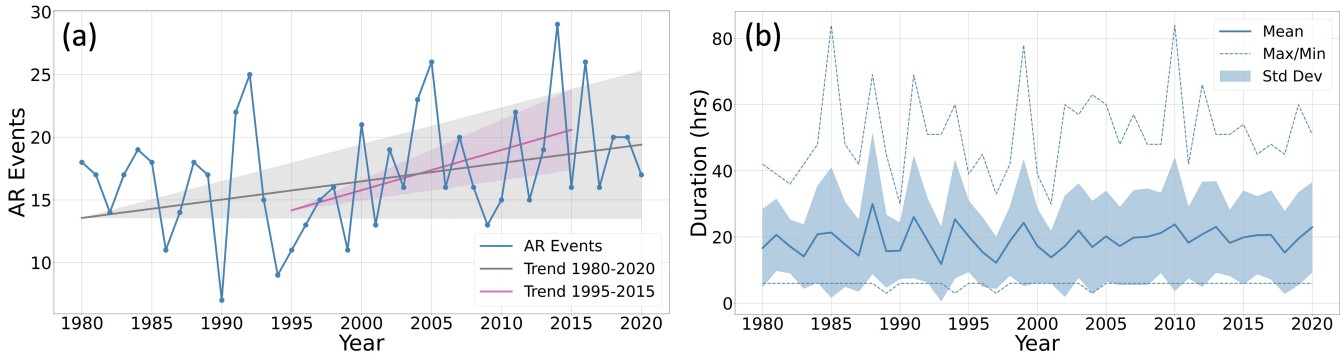

**Figure 3.** (a) The number of AR events making landfall over the Amundsen Sea Embayment and Marie Byrd Land each year (blue), with the 1980 to 2020 trend and standard error (grey) and the 1995 to 2015 trend error (pink). (b) the mean duration of AR events each year (thick blue line), along with the maximum and minimum duration of events (dotted lines) and the standard deviation from the mean (blue shading).

On average, there are $17 \pm 5$ AR events over the Amundsen Sea Embayment and Marie Byrd Land each year (Fig. 3). From 1980 to 2020, there is a positive trend in AR events of $+0.12 \pm 0.06$ events per year squared ($p = 0.055$), with a marked trend of $+0.32 \pm 0.16$ events per year squared ($p = 0.059$) from 1995 to 2015. The mean duration of AR events in this region is $16.2 \pm 13.9$ hrs, which indicates there is large variability in the duration of AR events. Throughout the study period, there were many short events that passed the AR detection threshold for only 3 hrs, while the longest detected durations after AR landfall occurred in 1985 and 2010, lasting 84 hrs. There is no statistically significant correlation between the number of AR events per year and their maximum, or mean, duration. Furthermore, there is no statistically significant seasonality to the number of AR events nor their duration over the Amundsen Sea Embayment and Marie Byrd Land.

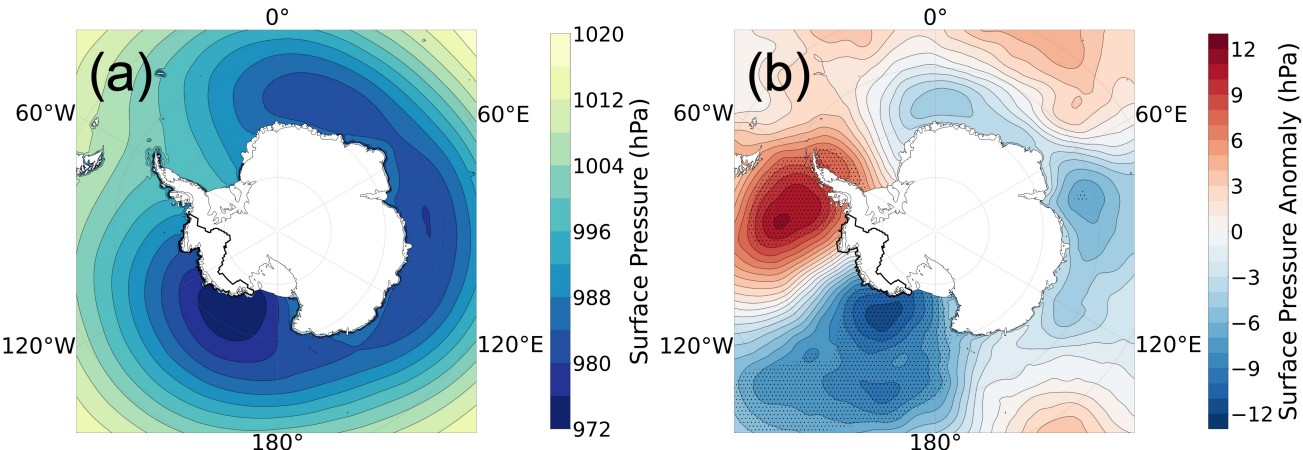

**Figure 4.** MERRA-2 surface pressure (a) and surface pressure anomaly (b) composites during AR landfall in the region of interest (the Amundsen Sea Embayment and Marie Byrd Land), which is outlined in black. Stippling on the surface pressure anomaly indicates regions of statistically significant anomaly, where the anomaly exceeds one half of the standard deviation from the mean surface pressure.

From the average surface pressure maps during AR events, we can identify that ARs in this region are driven by large-scale synoptic patterns involving the coupling of a blocking high and a low pressure system (Fig. 4). In the surface pressure composite of all AR events on TG, we find that AR events are associated with a climatological low–pressure system, commonly referred to as the Amundsen Sea Low, located west of TG (Raphael et al., 2016). There is also a high pressure ridge extending

southward from South America towards the Antarctic Peninsula. In the surface pressure anomaly composite, we see statistically significant high pressure anomalies over the Antarctic Peninsula, signaling the presence of a blocking high during AR landfall on TG. We also observe statistically significant low pressure anomalies in the region of the Amundsen Sea Low, west of TG. The blocking high serves to inhibit the zonal flow of low–pressure systems and directs circulation towards the Amundsen Sea Embayment and Marie Byrd Land. Furthermore, the surface pressure anomaly composite shows a zonal wave three-like

structure of alternating high and low pressure anomalies surrounding the AIS (Raphael, 2004).

To further examine the impacts of AR landfalls on TG surface conditions, we calculate the change in surface temperatures on Thwaites Eastern Ice Shelf during AR events (Fig. 5). To do this, we take the difference between the mean MERRA-2 2 m temperature 24 hrs before landfall, and the mean 2 m temperature 24 hrs after landfall. AR events are associated with a temperature increase of 1.4 K (first quartile) to 7.1 K (third quartile), with median 3.8 K, over Thwaites Eastern Ice Shelf over

all seasons. In austral summer (December-January-February), the median temperature increase is the smallest at 1.5 K. In fall



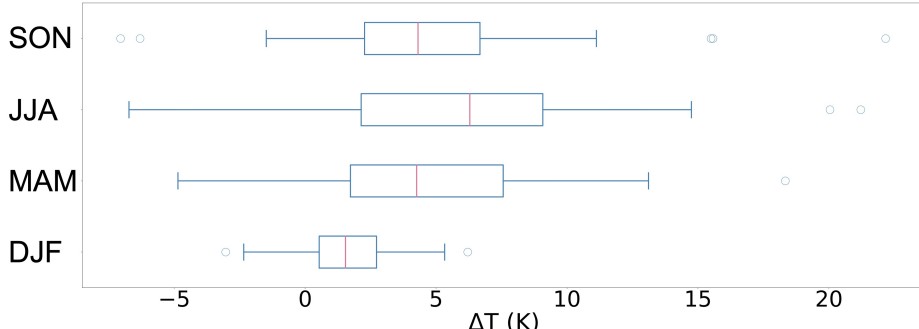

**Figure 5.** MERRA-2 2 m temperature difference by season between the average temperature 24 hours before AR landfall and the average temperature 24 hours starting at AR landfall (post– minus pre–event) from MERRA-2 reanalysis. Starting from the bottom, austral summer (December–January–February or DJF), fall (March–April–May or MAM), winter (June–July–August or JJA), and spring (September–October–November or SON). From left to right on each box plot: outliers, minimum, first quartile, median, third quartile, maximum, outliers.

(March-April-May), winter (June-July-August), and spring (September-October-November), the median temperature increases associated with AR landfall are 4.3 K, 6.3 K, and 4.3 K, respectively. The increase in surface temperatures is associated with temperature advection and increased downwelling longwave radiation during AR landfall (Wille et al., 2019). Here we see the largest increases in temperature associated with the landfall of winter AR events. There are many more summer events where

2 m temperatures exceed the melting point of 273.15 K (6 events in 1980-2020) than in fall (2 events), winter (1 event), and spring (0 events). While winter AR events are associated with the largest median and maximum temperature increases during AR landfall, they are also associated with the largest standard standard deviation in temperature changes (±5.0 K), and some winter events exhibit a decrease in temperature from the mean 24 hrs before landfall to the mean 24 hrs after landfall.

### 3.2   Case Study: Atmospheric River Surface Impacts on Thwaites Glacier

The geometry and orientation of TG render it highly susceptible to synoptic flow-induced snow storms caused by ocean air masses that are driven from the Southern Ocean into the ASE. When air masses rise up the slope of TG, the water vapor in marine air masses condenses, leading to orographic intensification of precipitation. With two AMIGOS installed on Thwaites Eastern Ice Shelf in January 2020, we are now able to examine unique in-situ atmospheric conditions during AR landfalls on Thwaites Eastern Ice Shelf. Case studies such as the February 2020 family of ARs can bring insight into the hydro–climate

dynamics responsible for West Antarctic AR climatologies and quantify examples of SMB impacts.

In February 2020, a family AR event occurred over TG over a period of 5 days, involving the landfall of 3 distinct ARs in rapid succession. The event began on February 2, when the first AR made landfall on the western flank of the glacier for 6 hrs (Fig. 6a, d). A second AR made landfall shortly after, from February 3 to 5, propagating from west to east across TG (Fig. 6b, e). A third AR made landfall from February 7 to 8, on the eastern flank of TG (Fig. 6c, f). This AR family event

was associated with a persistent high pressure ridge extending from South America to the Antarctic Peninsula, coupled with a





broad low pressure system northwest of TG. This blocking high acted to keep the low pressure in place, preventing its zonal migration.

During this time, several distinct sea–level pressure minima (nodes) developed within the broader surface low–pressure system, which were associated with the passage of short–wave troughs in the middle troposphere (Fig. 6a–c). Short–wave

troughs promote enhanced divergence in the middle troposphere and the subsequent development of localized sea–level pressure minima. These nodes facilitated the formation of ARs along the eastern edge of the broader surface low by enhancing the meridional transport of marine air masses southward. In particular, the first AR is driven by two nodes embedded within the broader surface low–pressure system, L1 and L2, which were associated with short–wave troughs T1 and T2. L1 and T1 subsequently dissipate, with the second AR driven by L2 and L3. The development of L3 can be tied to a separate short–wave

trough (T3) that propagated around the broader upper–level trough prior to the second AR. The third AR is driven exclusively by L4, which is associated with another short-wave trough (T4) that can be tracked from off the coast of east Antarctica on February 2 to TG on February 7. Coincident with this family of ARs, a persistent ridge of high–pressure, known as a blocking high, forms downstream of the broader surface low–pressure system. The coupling of the high– and low–pressure systems channels each AR into the Amundsen Sea Embayment and onto TG, leading all 3 ARs to make landfall at the same location.

The release of latent heat due to condensation and deposition within each AR amplifies the mid-tropospheric geopotential height anomalies, reinforcing the blocking high in a positive feedback cycle. This evolution permits multiple AR events to occur in rapid succession, as seen in this case study of the February 2020 AR family event.

On Thwaites Eastern Ice Shelf, we observe distinct signals in the AMIGOS surface pressure, as well as near-surface air temperature, wind direction and wind speed during the AR family event (Fig. 7). From the onset of the first AR, near-surface

temperatures increase to the melting point, and remain consistently at or above the melting point in observations and ERA5 (MERRA-2 remains below the melting point) for the entire duration of the AR family event. The observed surface pressure repeatedly drops during the AR events, indicating the onset of new low pressure system nodes associated with and reinforcing each AR. The wind direction experiences rapid changes before the first AR event, and after the third, and shifts to the north during the events. This reflects the synoptic pattern of northerly flow onto TG between the low pressure system and blocking

high. The wind speed accelerates during the first and second events, then drops and rises once more during the third AR landfall.

Next, we examine the impacts of the February 2020 AR family event on the SMB of Thwaites Eastern Ice Shelf (Fig. 8). From our SNOWPACK firn modeling analysis, which converted observed snow heights into accumulation associated with the AR family event, we find a total accumulation of 110 kg m$^{-2}$ at Cavity Camp and 97 kg m$^{-2}$ at Channel Camp. A rapid

increase in accumulation occurs at the onset of the first AR, and continues through the third AR. During this event, Cavity Camp exhibits earlier and higher total accumulation than Channel Camp, possibly due to local topographical differences between the two sites on Thwaites Eastern Ice Shelf. The SNOWPACK reconstructed accumulation at each site is very similar when using MERRA-2 or ERA5 radiative forcing.

When we compare the SNOWPACK reconstructed accumulation with the accumulation in MERRA-2 and ERA5 reanalyses,

we find that both reanalyses slightly underestimate the total accumulation during this time by 10–20%. MERRA-2 accumu-



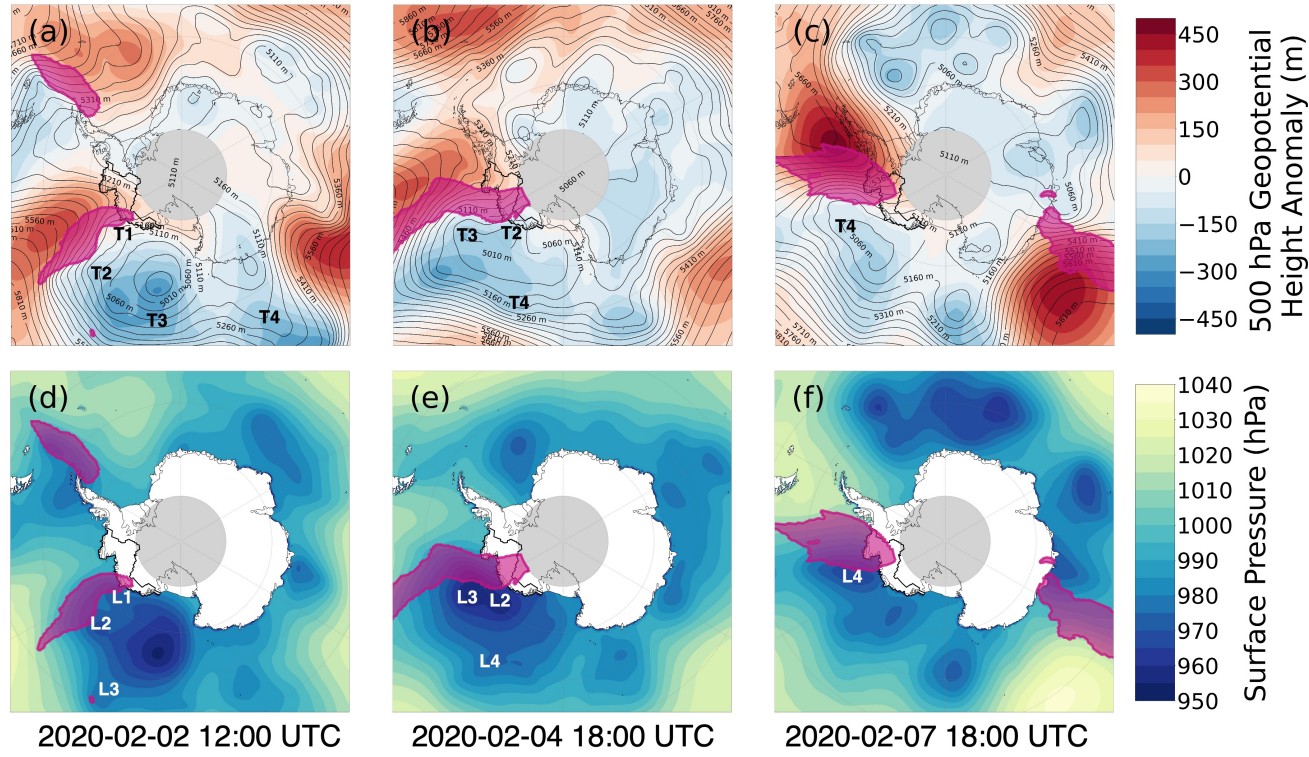

2020-02-02 12:00 UTC     2020-02-04 18:00 UTC     2020-02-07 18:00 UTC

**Figure 6.** 3 ARs make landfall on TG in short succession on TG on (a, d) February 2, (b, e) February 4, and (c, f) February 7, 2020. (Top row) MERRA-2 500 hPa geopotential heights (contours) and height anomaly (colors) during each AR landfall. (Bottom row) MERRA-2 surface pressure (colors) during each AR landfall. In each panel, the AR detection catalogue by Wille et al. (2021) indicates the spatial footprint of the ARs in pink. The location of short–wave troughs (geopotential height) and associated nodes in the low–pressure system (surface pressure) are indicated in each panel. Trough 1 (T1) drives Low 1 (L1), T2–L2, T3–L3, and T4–L4.

lation is 88 kg m$^{-2}$, and ERA5 accumulation is 87 kg m$^{-2}$. The total precipitation in 2020 is 1005 kg m$^{-2}$ in MERRA-2 and 983 kg m$^{-2}$ in ERA5, so this AR family event represents 9% of the total annual precipitation in both reanalysis products. While the snow height observations at Cavity Camp and Channel Camp and represent point locations, the accumulation in the reanalyses represents a grid-cell average. Therefore, the reanalyses may partly underestimate local accumulation on Thwaites

Eastern Ice Shelf, particularly during extreme events, due to the difference in spatial resolution.

     Based on the SNOWPACK analysis, we also find small amounts of surface melting associated with the AR family event. Alongside higher near-surface temperatures in ERA5 compared to MERRA-2 during the event, we also find slightly higher amounts and longer duration of surface melt when we use ERA5 radiative forcing. When using MERRA-2 radiative forcing, we find 2 kg m$^{-2}$ of surface melt at Cavity Camp and 1 kg m$^{-2}$ of total melt at Channel Camp. When using ERA5 radiative

forcing, we find 5 kg m$^{-2}$ at Cavity Camp and 3 kg m$^{-2}$ of total melt at Channel Camp. We find that the melt occurs for 34 hrs spread across 6 days at Cavity Camp with MERRA-2 radiative forcing and for 59 hrs spread across 11 days with ERA5





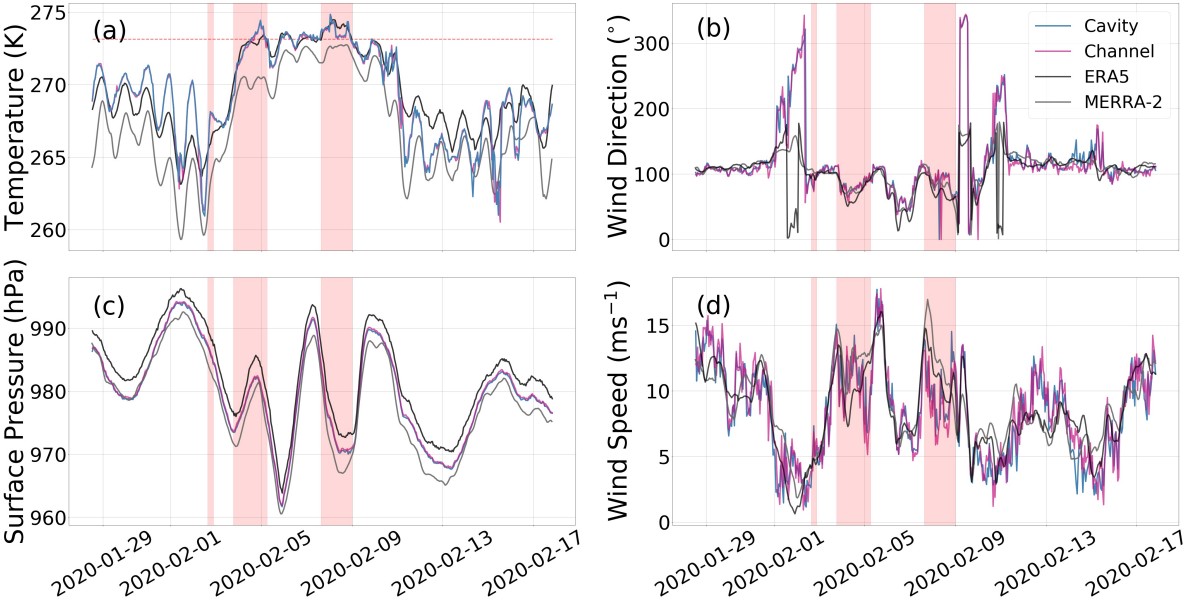

**Figure 7.** Observations of (a) 2 m temperature, (b) wind direction, (c) surface pressure, and (d) wind speed during the AR family event from AMIGOS at Cavity Camp (blue) and Channel Camp (pink) sites, and MERRA-2 (grey) and ERA5 (black) reanalyses. Pink shading indicates the timing of the AR events, based on the Wille et al. (2019) AR detection algorithm.

radiative forcing. At Channel Camp, the melt occurs for 26 hours over 3 days with MERRA-2 forcing and for 46 hours over 7 days with ERA5 forcing. Furthermore, thermistors placed 1.5 m above the battery box at the AMIGOS stations indicate a sub-surface increase in temperatures from -6 °C to the melting point coinciding with the second and third ARs in the AR

family event. The rise in firn temperatures indicates surface meltwater percolation through the firn (Supplementary Fig. B1). Overall, surface melt is nearly 2 orders of magnitude lower than the snowfall, indicating that the primary impact of this AR family event is to contribute snowfall to TG.

   Throughout the AR family event, accumulation is consistently higher at the Lower Thwaites GNSS site than at the Upper Thwaites GNSS site (Fig. 8). Lower Thwaites experiences 184 kg m$^{-2}$ total accumulation, and Upper Thwaites experiences

94 kg m$^{-2}$ total accumulation. The difference in accumulation between the two sites may indicate that Lower Thwaites experiences more orographic precipitation than Upper Thwaites (the sites are 129 km apart, and 304 m in vertical distance), and local topographical variation may contribute to the difference as well.

## 4   Discussion and Conclusions

ARs transport large amounts of moisture from the mid– to high–latitudes, leading to high snowfall events and elevated surface

temperatures when they make landfall over the AIS. By combining the large-scale climatology of AR events on the Amundsen Sea Embayment and Marie Byrd Land, and the case study of the surface impacts of an AR family event on TG, we aim to

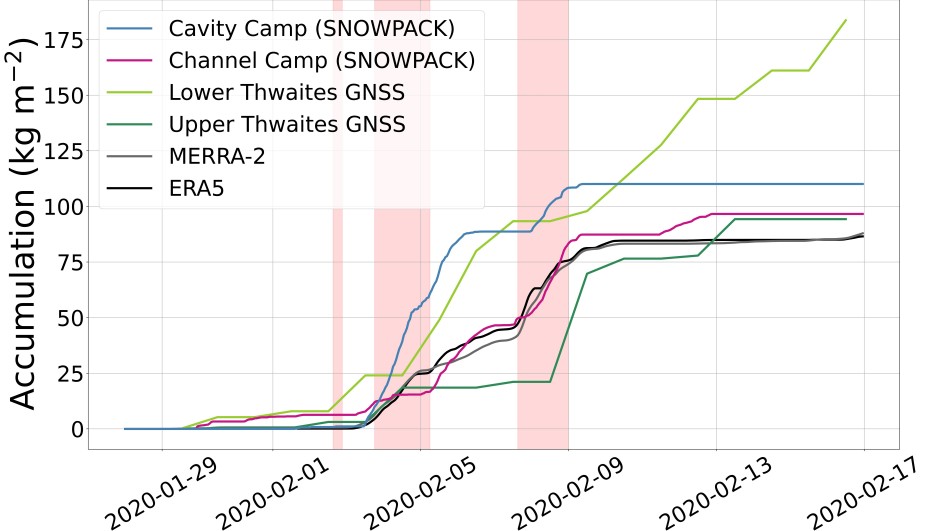

**Figure 8.** Reconstructed accumulation at Cavity Camp (blue) and Channel Camp (pink) sites on Thwaites Eastern Ice Shelf during the AR family event. AMIGOS observations of snow height and atmospheric conditions are used force SNOWPACK with radiation provided by MERRA-2. Light purple and dark purple lines show GNSS–IR–derived accumulation at Lower Thwaites and Upper Thwaites, respectively. We also show accumulation only from MERRA-2 (grey) and ERA5 (black). Pink shading indicates the timing of the AR events.

provide key insights on the nature and impacts of AR events in West Antarctica. In this study, we examine both the large–scale climatology of West Antarctic ARs using the vIVT AR catalogue developed by Wille et al. (2021) and reanalysis products, and the local effects of a case study AR family event in February 2020 on Thwaites Eastern Ice Shelf using AMIGOS weather
observations, the SNOWPACK firn model, GNSS-IR snow accumulation records, and temperature sensors in the firn.

Using the Wille2021AntarcticImpacts AR detection algorithm based on vIVT, we are able to diagnose the local climatology of ARs making landfall over the Amundsen Sea Embayment and Marie Byrd Land. This algorithm is uniquely adapted to focus on polar regions, where cold and dry air allow for less atmospheric moisture than in the mid–latitudes, where most AR detection algorithms currently focus (Rutz et al., 2019). However, the Wille et al. (2021) algorithm relies on only one
variable (vIVT) and one reanalysis (MERRA-2) to calculate the spatial footprints of ARs. Moreover, we currently do not have a method to validate the algorithm, as most global AR detection algorithms do not capture ARs so far south, and even this algorithm is limited to latitudes above 80 ° S. Future developments in Antarctic-specific AR detection algorithms should bring in observations and more than one atmospheric variable to classify the presence of an AR.

We find that AR events making landfall in the Amundsen Sea Embayment and Marie Byrd Land are driven by the coupling
of a blocking high over the Antarctic Peninsula with a low–pressure system known as the Amundsen Sea Low. While ARs are infrequent, they cause intense precipitation in short periods of time, and account for 11% of the annual surface accumulation in this region. There are 17 ± 5 AR events occurring over the Amundsen Sea Embayment and Marie Byrd Land each year, with a mean duration of 16 hrs. The total snowfall on TG and the top 10% highest snowfall days (1980-2015) exhibit strong





seasonality, with 2 to 3 times as much snowfall occurring in the fall, winter, and spring seasons when compared to the austral
summer (Maclennan and Lenaerts, 2021). AR events, however, which explain up to 10% of extreme snowfall events on the
Amundsen Sea Embayment and Marie Byrd Land (Wille et al., 2021), exhibit no statistically significant seasonality in the
number of AR events. This indicates that there are relatively more AR events making landfall over the AIS in the austral
summer when compared to the top 10% of snowfall events, or the total snowfall, than in the fall, winter, and spring seasons.
This difference may be explained to some extent by the greater baroclinicity of the atmosphere over the Southern Ocean and
AIS in the shoulder seasons, which may enhance the poleward transport of extra-tropical cyclones and the resulting occurrence
of winter storm events not associated to ARs (Van Den Broeke, 1998). ARs are rare, and since 90% of extreme snowfall days
are not associated with ARs detected by the Wille et al. (2021) vIVT algorithm, it is crucial to consider the role of non-AR
storm events in contributing to the SMB of TG as well.

    Global reanalyses, including MERRA-2 and ERA5, are employed at modest horizontal resolutions, and parameterize pre-
cipitation rates, likely leading to an underestimation of extreme local precipitation on the AIS. Furthermore, to attribute pre-
cipitation to ARs, we quantify precipitation directly beneath the spatiotemporal footprint of the AR as detected by the Wille
et al. (2021) algorithm, up to 24 hrs after landfall, which likely underestimates the true footprint of the AR. Therefore, in–situ
observations can provide more accurate information on the local amounts of snowfall associated with AR landfall over the
AIS. While short in time (lasting only 1.5 yrs from the date of installation), the AMIGOS weather observations are key to
diagnosing the surface impacts of the AR case study event in our study.

    We present a case study of an AR family which was marked by a series of 3 distinct ARs making landfall over TG in rapid
succession from February 2 to 8. During this AR family event, the AMIGOS indicate a large increase in snow height (0.3 m) on
Thwaites Eastern Ice Shelf. We use the SNOWPACK model to quantify the amount of mass added to the surface during the AR
family event by reconstructing accumulation based on AMIGOS–observed surface height changes. This approach represents
the essential role of firn modeling in converting observed surface height change to mass change on Antarctica. We find that
accumulation dominates the SMB impacts of the event, with 110 kg m$^{-2}$ and 97 kg m$^{-2}$ of snowfall at Cavity Camp and
Channel Camp, respectively. Combining SNOWPACK reconstructed accumulation on Thwaites Eastern Ice Shelf with GNSS-
IR snow accumulation records on TG adds a spatial component to the observed snowfall. During the case study event, we
see the highest accumulation at Lower Thwaites GNSS, followed by Cavity Camp and Channel Camp on Thwaites Eastern
Ice Shelf, and then Upper Thwaites GNSS. This is consistent with Maclennan and Lenaerts (2021), which found that based
on reanalyses MERRA-2 and ERA5, orographic precipitation on TG is highest in the region of steepest surface slopes, at
the lower part of the glacier. We find that using in–situ observations is necessary to capture the full surface impacts of AR
events, particularly to quantify accumulation, which is underestimated by reanalyses during the case study event. However,
among all of the in-situ observations (AMIGOS and GNSS-IR) and even with satellite altimetry (Adusumilli et al., 2021), the
problem persists of converting surface height change to mass change remains, amplifying the need for firn modeling to observe
glaciological changes on TG.

    From MERRA-2 reanalysis, we find that ARs over TG are associated with a 1.4 to 7.1 K temperature increase when they
make landfall over Thwaites Eastern Ice Shelf. The largest temperature increases during AR landfall are in the winter months





(June-July-August). There are steep temperature inversions present over the AIS in winter, so a disruption in the near-surface

atmospheric stability by AR events may be the driving factor for the large temperature increase (Phillpot and Zillman, 1970). The occurrence of summer ARs, however, is critically important for the SMB and firn health of the AIS. As surface-based temperature inversions are least developed in austral summer, the baseline surface temperatures before AR events are nearest the melting point in summer. When AR events make landfall and further raise surface temperatures through latent heat release and downwelling longwave radiation, they can cause surface melting on the AIS by raising temperatures to or above the

melting point (Wille et al., 2019). While ARs currently act as a net positive to the SMB of the AIS, future increases in surface temperatures may lead to larger-scale surface melting events, firn air depletion, and runoff.

From AMIGOS atmospheric observations on Thwaites Eastern Ice Shelf during the case study AR family event in February 2020, we observe a rapid increase in near-surface temperatures during the first 2 ARs, and temperatures remain at the melting point for the following 5 days. When comparing the observations to reanalyses, we find that MERRA-2 underestimates near-

surface temperatures during the event, while ERA5 shows more realistic temperatures. This highlights the need for in-situ observations — using only MERRA-2 reanalysis, we may not have seen the indication that the AR family caused surface melting on Thwaites Eastern Ice Shelf. When examining the long-term impacts of ARs on the surface and firn of the AIS, we cannot rely only on reanalysis, especially MERRA-2, to capture the full range of surface temperatures during AR events. To quantify surface melting during the AR family event using AMIGOS observations, we once again use SNOWPACK firn

modeling. By forcing SNOWPACK with observed near-surface temperatures, we find 2 to 5 kg m$^{-2}$ of melt at Cavity Camp and 1 to 3 kg m$^{-2}$ of melt at Channel Camp (when using MERRA-2 and ERA5 radiative forcing). Furthermore, AMIGOS temperature sensors 1 m below the surface indicate a sudden temperature rise due to meltwater percolation in the firn.

The greatest limitations to the in-situ weather observations are spatial and temporal coverage. The AMIGOS captured hourly atmospheric data on Thwaites Eastern Ice Shelf for only 1.5 yrs, after which data collection stopped due to high winds on

Thwaites Eastern Ice Shelf. These data provide a ground truth for surface conditions and impacts of weather events, including but not limited to ARs. GNSS-IR records of accumulation span the last two decades, providing an accumulation record that we can compare to the AMIGOS observations and reanalysis. Still, with limited temperature observations on Thwaites Eastern Ice Shelf, and no measurements of atmospheric radiative fluxes, we have extremely limited information on the state of the firn on the ice shelf and TG. Therefore, firn modeling is critical in quantifying both accumulation and melt during AR events, because

it can inform us about snow density and compaction rate, surface melt, snow redistribution, and sublimation — processes that all affect observed surface height change but are not easy to separate.

AR-driven surface melting will likely not lead to the destabilization of Thwaites Eastern Ice Shelf, particularly when compared to the pace and extent of current ocean-induced basal melting of the ice shelf. As seen during the case study event, snowfall dominates AR surface impacts by two orders of magnitude when compared to melt, and in general surface temper-

atures rarely reach the melting point on Thwaites Eastern Ice Shelf. However, ARs represent an important contributor to the mass balance of TG. The snow accumulation and range near–surface temperatures associated with these extreme events have the largest impacts on the SMB and may affect feedbacks with ice dynamics on TG. If these extremes are amplified in the future climate, such as an increase in the frequency and/or intensity of ARs, we may observe more frequent surface melting in



West Antarctica. This could seriously impact ice shelf stability, the ability of the firn layer to absorb melt water, and ultimately
the mass balance of the WAIS. In turn, changes in ice dynamics and surface height on TG may impact the location and intensity
of AR-attributed orographic precipitation (Christian et al., 2022). Finally, surface melt is not only relevant for ice shelf stability
and glacier mass balance — it matters for firn temperatures and air content, and glaciological observations of mass change of
TG. Melt changes thermal structure of the firn, and thus there are hidden impacts of surface melt that are not visible at the
surface. This highlights the need to examine the representation of ARs in climate models and how their intensity, frequency,
and SMB impacts may change in the future.

While AR events occur slightly more frequently over the Antarctic Peninsula and Dronning Maud Land than over the
Amundsen Sea Embayment and Marie Byrd Land (Wille et al., 2021), the vulnerability of the latter region to ocean-induced
ice mass loss and ice sheet instability amplifies the importance of quantifying accumulation and the interannual variability
of AR events as a compensation mechanism for the mass loss. The longer the duration of the AR event, the more intense
its impacts, as demonstrated in the case study of the extreme AR family event in February 2020. The baseline atmospheric
temperature, and the extent to which the AR raises surface temperatures, plays a key role in AR-driven surface melting in
West Antarctica (Wille et al., 2019). While West Antarctica currently experiences minimal surface melting, most of which
is absorbed by the firn, future-climate scenarios could exhibit more widespread surface melting in West Antarctica. Future
warming in Antarctica many create a situation similar to the present–day Greenland Ice Sheet, where melting occurs at lower
elevations and snowfall occurs at higher elevations during AR events, with a potential increase in AR-driven rainfall as well
(Mattingly et al., 2018, 2020). Combined with the dynamic mass loss of West Antarctica, changes in AR–SMB impacts may
affect the stability and mass balance of the WAIS. The observations of the case study AR event underline the importance of AR
representation in modeled future climate scenarios, and analysis of how the frequency, duration, and intensity of AR events,
and the occurrence of AR family events, may change in the future.

*Code and data availability.* The AMIGOS weather and snow height data are available from USAP-DC at https://www.usap-dc.org/view/
dataset/601552 and https://www.usap-dc.org/view/dataset/601549. MERRA-2 data are available through the Goddard Earth Sciences Data
and Information Services Center at https://disc.gsfc.nasa.gov/datasets/M2T1NXLFO_5.12.4/summary. ERA5 data are available through
the Copernicus programme Climate Data Store at https://cds.climate.copernicus.eu/cdsapp#!/dataset/reanalysis-era5-single-levels?tab=form.
The Wille vIVT catalogue is available through the ARTMIP database, Doi:10.5065/D6R78D1M. MeteoIO and SNOWPACK are soft-
ware published under the GNU LGPLv3 license by the WSL Institute for Snow and Avalanche Research SLF, Davos, Switzerland at
https://gitlabext.wsl.ch/snow-models. The repository used to develop the versions of MeteoIO and SNOWPACK used in this study can
be accessed at https://github.com/snowpack-model/snowpack with the exact version corresponding to commit 149c586. GNSS-IR snow ac-
cumulation time series is available through the University of Washington research works archive http://hdl.handle.net/1773/48595. Code
used to derive GNSS-IR accumulation history is available at https://github.com/hoffmaao/thwaites-gnss-ir.

**Appendix A**

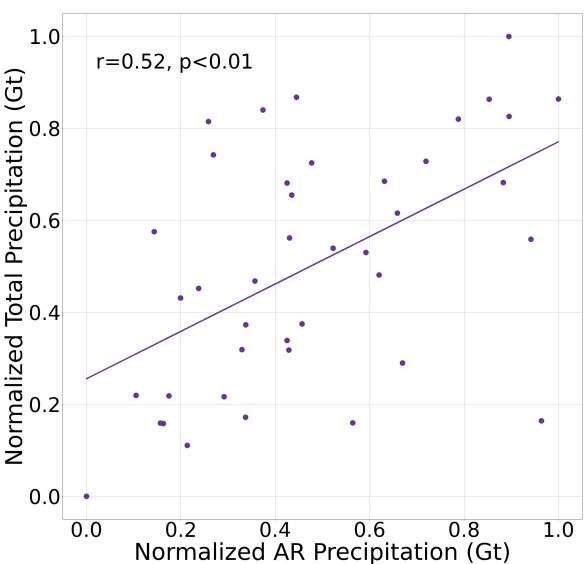

**Figure A1.** Correlation between normalized annual AR precipitation and the normalized annual total precipitation over the Amundsen Sea Embayment and Marie Byrd Land. The Pearson correlation coefficient $r = 0.52$ $(p < 0.01)$.

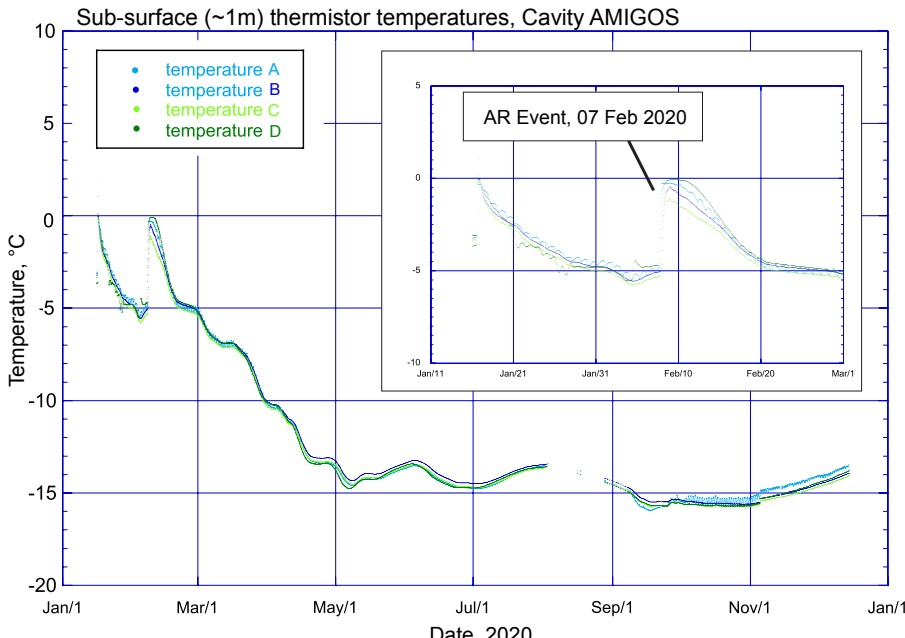

**Figure B1.** Firn temperature time series during the February 2020 AR event from 4 thermistors (a, b, c, and d), all placed 1 m below the surface at Cavity Camp.

*Author contributions.*

M. L. Maclennan led the data analysis and writing. J. T. M. Lenaerts assisted in data analysis and writing. C. A. Shields and J. D. Wille provided access to the atmospheric river detection catalogue and assisted in data analysis. A. C. Winters provided analysis of the synoptic meteorology during the case study event. A. O. Hoffman provided GNSS-IR time series of snow accumulation on Thwaites Glacier. N. Wever and M. Thompson–Munson performed SNOWPACK modeling of the firn, driven by AMIGOS observations. T. A. Scambos provided the AMIGOS data, and led the associated fieldwork with E. C. Pettit.

*Competing interests.* The authors declare that they have no conflict of interest.

*Acknowledgements.* This work is from the TARSAN project, a component of the International Thwaites Glacier Collaboration (ITGC). Support from National Science Foundation (NSF: Grant 1929991) and Natural Environment Research Council (NERC: Grant NE/S006419/1). Logistics provided by NSF-U.S. Antarctic Program and NERC-British Antarctic Survey. ITGC Contribution No. ITGC-073.

We would like to recognize Tim White and Bruce Wallin for building and programming the AMIGOS-III units. We acknowledge the Colorado Space Grant at the University of Colorado for the undergraduate team that first integrated the new sensors to the AMIGOS-III



system. Ronald Ross and Ted Scambos developed the earliest version of the AMIGOS concept. Finally, we thank the TARSAN field team for the installation of the AMIGOS on Thwaites Eastern Ice Shelf, including Erin Pettit, Christian Wild, Karen Alley, Gabriela Collao–

Barrios, Atsuhiro Muto, Martin Truffer, Cecelia Mortenson, Blair Fyffe, and Dale Pomraning. We also acknowledge Antarctic Support Contractors/Leidos and Ken Borek Air.

M. L. Maclennan acknowledges support from NASA FINESST grant 80NSSC21K1610. C. A. Shields acknowledges the U.S. Department of Energy, Office of Science, Office of Biological & Environmental Research (BER), Regional and Global Model Analysis (RGMA) component of the Earth and Environmental System Modeling Program under Award Number DE-SC0022070 and National Science Foundation

(NSF) IA 1947282. A. O. Hoffman acknowledges support from NSF OPP 1738934. M. Thompson-Munson was supported by the National Aeronautics and Space Administration Interdisciplinary Research in Earth Science Grant No. 80NSSC20K1727. A. C. Winters acknowledges support from start-up funding furnished by the University of Colorado Boulder. J. D. Wille acknowledges support from the Agence Nationale de la Recherche projects ANR-20-CE01-0013 (ARCA).



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
