# Peer review of "Climatology and Surface Impacts of Atmospheric Rivers on West Antarctica"

_The Cryosphere, 2022_

## Author Comment (AC1)

**Climatology and Surface Impacts of Atmospheric Rivers on West Antarctica**

Michelle L. Maclennan, Jan T. M. Lenaerts, Christine A. Shields, Andrew O. Hoffman, Nander Wever, Megan Thompson-Munson, Andrew C. Winters, Erin C. Pettit, Theodore A. Scambos, and Jonathan D. Wille

**REVIEWER COMMENT #1:**

Review "Climatology and Surface Impacts of Atmospheric Rivers on West Antarctica" by Michelle Maclennan and co-authors.

This manuscript investigates the climatological conditions and the surface impacts of atmospheric rivers (ARs) in West Antarctica. The author first uses reanalysis model output (MERRA-2) in combination with an AR detection tool to examine the contribution of ARs in this region from 1980 to 2020. Then for a more detailed and smaller scale perspective the authors present a case study of three successive ARs on Thwaites Glacier in February 2020, for which they use reanalysis data, in-situ measurements and a firn model. Finally, the authors discuss how ARs may change in a future climate.

The manuscript is well written with clear figures. It is an interesting and relevant study within the scope of TC. The idea and methods are not completely new, it builds on existing knowledge from ARs in Antarctica and previous firn modeling efforts. By combining large scale model output and in-situ measurements, the results are a useful contribution for understanding the climatology and impacts of atmospheric rivers in West Antarctica. Despite being a topic of interest, there are some minor aspects especially regarding the contribution/purpose, goals stated in introduction, methodology and results that might be better represented. I elaborate on this in the comments below, which follow the order of the manuscript.

**The authors would like to thank Sanne Veldhuijsen for their feedback and for providing insightful recommendations on improving the motivation of the study and the clarity of the text. We have responded to the reviewer's comments as follows, with a particular focus on revising the last two paragraphs of the introduction and adding context to the introduction and discussion sections with references to previous studies on Antarctic ARs. Responses are written in bold, and excerpts from the manuscript are *italicized*. Changes to the text are *italicized and in blue*. Line numbers refer to track changes in the revised manuscript.**

General comments/questions:

1. Contribution/purpose of this study: an elaborate introduction about AIS mass balance and atmospheric rivers is given. However, the articulation of the purpose and contribution of this study in the introduction can be improved. Articulate more clearly what is new in this study compared to previous work, the added value of this

study, what is already known about ARs this region (What have Willle et al. 2019 & 2021 found about ARs in West-Antarctica, e.g. how many per year/trend)? Also the reason why is chosen for this region (Lines 123-127) would be more suitable for this part of the introduction.

**We have revised the last two paragraphs of the introduction (starting on line 58) to emphasize the motivation for our study and its contribution in the context of Antarctic ARs research. We now explain that based on previous studies, the spatial variability of extreme precipitation associated with ARs over West Antarctica is poorly understood and fails to capture the local accumulation associated with AR events. We highlight that our analysis of a case study event provides key indications of small-scale spatial variability in AR-driven accumulation and surface melting on Thwaites Eastern Ice Shelf. Finally, we now also emphasize that placing this case study within the broader context of the climatology of West Antarctic ARs enables us to better understand the characteristics and impacts of ARs on the surface mass balance.**

**In line 203 in the results, we now discuss how the trend in AR events we found is consistent with the trend found by Wille et al. (2021):**

***From 1980 to 2020, there is a positive trend in AR events of +0.12 +/- 0.06 events per year squared (p = 0.055), similar to the results from Wille et al. (2021), which also showed an increasing trend in AR frequency from 1980 to 2018 over the WAIS region.***

**In response to comments from another reviewer, we have revised Data and Methods section 2.2 "Reanalysis Products: MERRA-2 and ERA5" by sticking specifically to information about the reanalyses and how they are used, and removing the text on lines 123-127, which is background information already mentioned in the introduction.**

2. Contrasting impacts on SMB: In Lines 76-78 you state that ARs have contrasting impacts on SMB, and that it is therefore important to study them from both large-scale climatological perspective and a case study. With contrasting impacts on SMB, I understand that you mean snowfall, melt or temperature? However, the melting (and temperature) part is not studied from the large-scale climatological perspective. Nevertheless, melt could be important on e.g. Abbot ice shelf. I think it would be good to explain in the introduction that and why the focus of the large-scale climatological perspective is on precipitation. This is probably also why you use the vIVT detection algorithm.

**We have revised the last two paragraphs of the introduction to improve clarity in response to several reviewer comments, including the sentence in question here. The paragraph starting on line 58 now explains the complex impacts of ARs on the surface mass balance of the Antarctic Ice Sheet. The paragraph starting on line 70 explains the motivation for our climatology and case study and highlights the gap in prior research that our study addresses. We do not aim to quantify large-scale AR-driven surface melting in this study, which has already been studied in Wille et al. (2019), and is much smaller in magnitude than AR-driven precipitation (this is mentioned in the introduction on line 67). Here, we emphasize that analyzing the large-scale climatology of ARs themselves improves our understanding of their characteristics and surface mass balance impacts, with a focus on quantifying AR-attributed precipitation. In section 2.3, we explain the motivation for using the vIVT detection algorithm - which is indeed better suited to quantifying precipitation than the IWV detection algorithm (Wille et al., 2021).**

3. Lines 85-86 "Finally, we discuss the results in the context of how ARs contribute to the present mass balance of the AIS and how their frequency and precipitation may change in future climate scenarios." I don't see where the future frequency and precipitation is discussed in the manuscript? You do discuss a potential increase in melt related to AR events. Perhaps use: "Responses and impacts of atmospheric rivers to climate change by Payne et al. (2020)", and the fact that there is an ongoing increase over time of current AR events, which might continue (Fig. 3a).

**The majority of the discussion on the future of Antarctic ARs indeed focuses on increased surface melting. As mentioned above, we revised the last two paragraphs of the introduction, including the text in question. To reflect this critical component of the Discussion, we rewrote the last sentence of the introduction as follows, to focus more on how the impacts of ARs may change in the future (line 80):**

*Finally, we discuss how ARs contribute to the present mass balance of the WAIS, which improves our understanding of how their impacts may change in future climate scenarios.*

4. The discussion is strong and very interesting, one thing that might be added is some comparison to previous findings about ARs on the WAIS, which is mentioned above as well. (E.g. Wille et al. 2019 & 2021).

**Thank you. We have added several references to the discussion section to discuss how the findings from this study compare to Scott et al. (2019), Wille et al. (2021), and Adusumilli et al. (2021).**

**Starting on line 325:**

*This pressure anomaly pattern is similar to the Pacific South-American patterns identified by Scott et al. (2019) as drivers of marine air intrusions and West Antarctic surface melting, and consistent with geopotential height anomalies identified by Adusumilli et al. (2021) during WAIS AR events in 2019. While ARs are infrequent, they cause intense precipitation in short periods of time, and account for 11% of the annual surface accumulation in this region, consistent with Wille et al. (2021).*

5. Lines 373-375: "Limited by 1.5 years of in-situ data." I wonder why you only look at 1 AR family event, while there are multiple AR events each year?

**In this paper we focus on an exceptional AR family event as a case study that occurred during the unique period when automatic weather stations recorded meteorological conditions on Thwaites Eastern Ice Shelf. We perform detailed analysis of both the specific atmospheric conditions that led to the event and the surface mass balance impacts that the family event had. The motivation to use a case study event here is that it allows us to closely examine the drivers and effects of a particular event, within the broader context of the climatology of West Antarctic ARs. On average, there are 9 +/- 3 AR events on Thwaites Glacier each year. In 2020, there were 8 ARs that made landfall over Thwaites Glacier, 3 of which corresponded to the February 2020 family event. None of the remaining 5 ARs were part of family events. Furthermore, wind speed sensors on the automatic weather stations experienced riming starting in June 2020, meaning the most complete record of observations is from January - June 2020. That is why we selected the February 2020 AR family event for the case study.**

Specific comments/questions:

1. Lines 63: Why is this unique for Antarctic ARs? Is this not the same for Greenland ARs?

**This is correct, Greenland ARs also have multiple effects on the SMB. Here, the comparison was aimed at highlighting how Antarctic ARs are different from midlatitude ARs. To avoid confusion, we have removed the first sentence of the paragraph and revised the following sentences to include the statement that Antarctic ARs can have multiple effects on the SMB, without calling them "unique" (line 58).**

* While ARs make landfall up to 14% of the time in the mid-latitudes (~50 days per year), they are comparatively rare over the AIS, making landfall only 1% of the time (or ~3 days per year, Rutz et al., 2019; Wille et al., 2021). Although they occur infrequently, Antarctic ARs can have multiple, contrasting*

*impacts on the SMB. ARs cause intense precipitation when they make landfall over the AIS, because they carry so much moisture.*

2. Lines 66: They carry much moisture, but does the fact that the AIS is a desert not also play a role in the importance of ARs?

**Yes, Antarctica is a desert, meaning that individual AR events can contribute significantly to local SMB (Gorodetskaya et al., 2014), particularly over East Antarctica. However, polar ARs are generally less moist than mid-latitude ARs, meaning the moisture threshold for Antarctic ARs is lower than the threshold in the mid-latitudes, so it is all relative (Wille et al., 2021). Additionally, Wille et al. (2021) showed that ARs contribute more of the annual SMB in East Antarctica than in West Antarctica - which experiences more total snowfall than East Antarctica (Lenaerts et al., 2019).**

**Gorodetskaya, I. V., Tsukernik, M., Claes, K., Ralph, M. F., Neff, W. D., and Van Lipzig, N. P. M. (2014): The role of atmospheric rivers in anomalous snow accumulation in East Antarctica, Geophysical Research Letters, doi: 10.1002/2014GL060881**

**Lenaerts, J. T. M., Medley, B., Broeke, M. R., and Wouters, B. (2019): Observing and Modeling Ice Sheet Surface Mass Balance, Reviews of Geophysics, doi: 10.1029/2018RG000622**

**Wille, J. D., Favier, V., Gorodetskaya, I. V., Agosta, C., Kittel, C., Beeman, J. C., Jourdain, N. C., Lenaerts, J. T. M., and Codron, F. (2021): Antarctic atmospheric river climatology and precipitation impacts, Journal of Geophysical Research: Atmospheres, doi: 10.1029/2020JD033788**

3. Line 72: The study of Neff et al 2014 is about Greenland.

**We have moved the Neff et al. (2014) reference to the following sentence in the paragraph (line 66). We have added Wille et al. (2019, 2021) as supporting references to the following sentence as well (in response to the next comment).**

*This causes surface melting in coastal Antarctica, particularly on the Antarctic Peninsula, which can lead to runoff and/or deplete the ability of the firn to store future meltwater (Wille et al., 2019; ). Unlike on the Greenland Ice Sheet (Neff et al., 2014; Mattingly et al., 2018), ARs act to increase Antarctic SMB, as they cause significantly more snowfall than surface melting (Wille et al., 2019; Wille et al., 2021).*

4. Line 73: "ARs act to increase Antarctic SMB, as they cause significantly more snowfall than surface melting". Can you give a reference for this statement?

**Yes, we have added Wille et al., (2019, 2021) to support the statement, please see previous comment.**

5. Lines 81-82: "to provide key insights on in-situ conditions" this can be rephrased. In-situ is often only used to describe the way a measurement is taken, maybe replace by local conditions.

**We have rewritten this section in response to Reviewer Comment #4, and the phrase in question has been revised as follows (line 77):**

*Then, we use in–situ observations and a firn model to examine the specific impacts of a series of three successive ARs that made landfall on TG in February 2020, as well as the ability of reanalyses to reproduce those observations. Our analysis provides key indications of small-scale spatial variability in AR-driven accumulation and surface melting on TG, within the broader context of the climatology of ARs in the region.*

6. Line 93: basal channel?

**Yes, a basal channel - we have added this in (line 89):**

*Cavity Camp is located on a flat part of Thwaites Eastern Ice Shelf, whereas Channel Camp sits within the surface expression of a basal melt channel (Alley et al., 2016).*

**Alley, K.E., Scambos, T.A., Siegfried, M.R. and Fricker, H.A. (2016). Impacts of warm water on Antarctic ice shelf stability through basal channel formation. Nature Geoscience, doi: 10.1038/ngeo2675**

7. Section 2.2: Is there a reason why you chose MERRA-2 instead of ERA-5? Perhaps you can add that both reanalysis products give similar results in Wille et al. 2021.

**We have added a sentence to section 2.2 describing why we choose to use MERRA-2 (line 114):**

*We primarily use MERRA-2 analyze the large-scale synoptics and impacts of AR events in West Antarctica, as MERRA-2 explicitly represents ice sheet hydrological and energy budgets and compares best to ice core records of snow accumulation in Antarctica among multiple reanalyses (Gelaro et al., 2017; Medley and Thomas, 2019).*

8. Line 135: Not over the AIS but over the WAIS.

**We have changed "AIS" to "WAIS" (line 133):**

*We use a polar-specific AR detection algorithm produced by Wille et al. (2021) to identify the occurrence and landfall of ARs over the WAIS.*

9. Line 153: Actually, you use three different approaches, also the GNSS measurements.

**We have eliminated this sentence, and the following sentence, to omit the discussion of reanalysis products in section 2.4 and focus on the SNOWPACK modeling (line 151). Thereby we have also deleted our mention of the number of approaches.**

*We determine the accumulation attributed to the case study AR event of February 2020 using two distinct approaches. In the first, we use precipitation from MERRA-2 and ERA5 reanalyses. In the second, We use observed snow height from the AMIGOS to force the firn model SNOWPACK to reconstruct accumulation during the AR case study event in February 2020.*

10. Section 2.4: I think it can be clarified how the firn modelling works. Perhaps add that snowfall is assumed to occur when measured snow height exceeds the modeled snow height. Strictly speaking, there can also be snowfall when the observed snow height remains stable e.g. if there is snowfall in combination with densification, sublimation or melt. The difference between the observed and modelled snow height is then added to the snowpack, which can be converted with the fresh snow density to accumulation.

**The reviewer indeed provides a clearer description of our use of the SNOWPACK model. We improved our description in the manuscript (section 2.4, starting on line 151):**

*We use observed snow height and temperature from the AMIGOS to force the firn model SNOWPACK (Lehning et al., 2002a, b) to reconstruct accumulation and surface melt during the AR case study event in February 2020. SNOWPACK is a physics-based, multi-layer firn model (Lehning et al., 2002a, b) which has been extensively applied in polar regions (Groot Zwaaftink et al., 2013; Steger et al., 2017; Van Wessem et al., 2021; Keenan et al., 2021). The model calculates snow compaction using an overburden formulation and solves the full surface energy balance to provide the upper boundary condition for solving the temperature equation and calculating melt. When snow accumulates, fresh accumulation density is calculated as a function of meteorological conditions, particularly wind and the presence of drifting snow (Keenan et al., 2021; Wever et al., 2022). We configure the SNOWPACK model here to derive snowfall from the observed snow height: if observed snow height exceeds the simulated snow height, the difference is interpreted as snowfall, when relative humidity, air temperature and snow surface temperature meet snowfall conditions (Lehning et al., 1999; Wever et al., 2015). This is then combined with fresh accumulation density, to convert to accumulated mass.*

11. Line 190-191: Perhaps start the results section with one sentence describing the kind of results you are going to show in Figure 2, as an introduction to the reader. Also refer to panels of figures if that is the case, so Figure 2a e.g in Line 193.

**We have added a sentence to the beginning of the paragraph to introduce AR frequency and how it is calculated. We moved the reference to Fig. 2 in the first sentence of the results to the third sentence, where we discuss local AR frequencies. We have added references to specific panels of Fig. 2 (a, b, and c) in the revised text below (line 195):**

*To determine the frequency of ARs over the Amundsen Sea Embayment and Marie Byrd Land region, we divide the number of AR times by the total time from 1980 to 2020. Our analyses show that ARs exhibit a total frequency of 3.2% over the whole region from 1980 to 2020 (i.e., there is an AR making landfall somewhere in the region 3.2% of the time, on average) (Fig. 2). This represents the total frequency of ARs over the region, calculated by dividing the number of AR times by the total time from 1980 to 2020. Within the region, localized AR frequencies range from 0.2 to 0.8% of the time, with the highest frequencies over the Abbot Ice Shelf and the Getz Ice Shelf (Fig. 2a). Integrated over the entire region, ARs contribute 59 +/- (one standard deviation) 24 Gt precipitation annually (out of 550 +/- 63 Gt total annual precipitation, Fig. 2b and c), and explain 28.7% of the interannual variability in precipitation (linear trends removed).*

12. Lines 190-191: the 3.2% in combination with the reference to figure 2 is a bit confusing as I don't see 3.2% in the figure. Also I suggest to first give the definition of frequency of ARs and then discuss the numbers.

    **Please see response to previous comment - we have added an introductory sentence with a definition of AR frequency, and moved the reference to Fig. 2a to a later sentence which discusses local AR frequencies over the region.**

13. Figure 3: Would it not also be interesting (and possible) to have a third graph with the amount of precipitation from ARs over time?

    **We have shown the annual mean contribution of ARs to precipitation over the Amundsen Sea Embayment and Marie Byrd Land spatially in Fig. 2b and c, and we describe the integrated annual mean precipitation in the first paragraph of the results. However, we agree that the time series of AR precipitation complements the time series of AR events in Fig. 3a. Therefore, we have added a Fig. 3c, which is a time series of AR-attributed precipitation by year from 1980-2015:**

[Figure]

14. Caption Figure 5: I suggest to add that this figure is about Thwaites Eastern Ice Shelf.

**We have added "Thwaites Eastern Ice Shelf" to the first sentence of the Fig. 5 caption.**

*MERRA-2 2 m temperature difference by season between the average temperature 24 hours before AR landfall and the average temperature 24 hours starting at AR landfall (post- minus pre-event) from MERRA-2 reanalysis over Thwaites Eastern Ice Shelf.*

15. Line 228: The temperature decrease is not only in the winter right?

    **We assume that the comment is referring to this statement (line 231):**

    *Here we see the largest increases in temperature associated with the landfall of winter AR events.*

    **On average, we observe a temperature increase in all seasons; however, the largest temperature increases occur in the winter season.**

16. Line 228: perhaps omit: "from the mean 24 hrs before landfall to the mean 24 hrs after landfall." as this should already be clear.

    **We assume that this comment refers to line 225:**

    *To do this, we take the difference between the mean MERRA-2 2 m temperature 24 hrs before landfall, and the mean 2 m temperature 24 hrs after landfall.*

    **We will keep this sentence in the results because it is explicit about the method used to calculate the change in temperature during AR events, and we (the co-authors) are in agreement that this phrase is important for the clarity of the results.**

17. Caption Figure 6: Line 1 omit repetition of "on TG".

    **Thank you for pointing this out, we have removed the repetition of "on TG" from the first sentence of the Fig. 6 caption:**

    *3 ARs make landfall on TG in short succession  on (a, d) February 2, (b, e) February 4, and (c, f) February 7, 2020.*

18. Line 281: Should "different spatial resolution" not be "low spatial resolution"?

    **We have changed the wording to "lower spatial resolution" (line 288).**

    *While the snow height observations at Cavity Camp and Channel Camp and represent point locations, the accumulation in the reanalyses represents a grid-cell average. Therefore, the reanalyses may partly underestimate local accumulation on Thwaites Eastern Ice Shelf, particularly during extreme events, due to the lower spatial resolution.*

19. Line 281: Perhaps include in method section 2.4 that you calculate surface melt from the SNOWPACK model.

We added that the SNOWPACK model calculates the full surface energy balance to provide the upper boundary condition for the temperature equation and to calculate melt (line 154):

*The model calculates snow compaction using an overburden formulation and solves the full surface energy balance to provide the upper boundary condition for solving the temperature equation and calculating melt.*

20. Caption Figure 8: Line 2: atmospheric conditions are used "to" force SNOWPACK.

**We have added "to" to the sentence in the Fig. 8 caption:**

*AMIGOS observations of snow height and atmospheric conditions are used to force SNOWPACK with radiation provided by MERRA-2.*

21. Line 306: Improve the reference formatting.

**The reference has been corrected (line 316):**

*Using the Wille et al., (2021) AR detection algorithm based on vIVT, we are able to diagnose the local climatology of ARs making landfall over the Amundsen Sea Embayment and Marie Byrd Land.*

22. Line 320: From Wille et al. 2021 I understand that 10% of total snowfall comes from AR events. The percentage of extreme precipitation events explained by ARs depend on the threshold but is 10% lower in West Antarctica than East Antarctica (where it is 25-45%).

**Wille et al. (2021) states that in West Antarctica, approximately 10% of the total annual snowfall comes from AR events. In East Antarctica, 10-20% of the total annual snowfall comes from AR events. The percentage of extreme precipitation events explained by ARs at the 90th percentile is 25-35% in East Antarctica. Fig. 4a in Wille et al. (2021) shows that over West Antarctica, the percentage of extreme precipitation events explained by ARs at the 90th percentile is 10%. Wille et al. (2021) states that on average, ARs explain 10% more extreme precipitation events in East Antarctica than in West Antarctica.**

23. Line 356-357: "As surface-based temperature inversions are least developed in austral summer, the baseline surface temperatures before AR events are nearest the melting point in summer." And also because it is simply warmer in summer?

**Yes - we have added this in to the sentence (line 369):**

*As surface-based temperature inversions are least developed in austral summer, and air temperature is higher than in other seasons, the baseline surface temperatures before AR events are nearest the melting point in summer.*

---

## Author Comment (AC2)

**Climatology and Surface Impacts of Atmospheric Rivers on West Antarctica**

Michelle L. Maclennan, Jan T. M. Lenaerts, Christine A. Shields, Andrew O. Hoffman, Nander Wever, Megan Thompson-Munson, Andrew C. Winters, Erin C. Pettit, Theodore A. Scambos, and Jonathan D. Wille

**REVIEWER COMMENT #2:**

Figure 3: I am not sure if the standard error is used in the right way here. Perhaps replace by the 95% confidence interval, as this is also easier to grasp.

**The standard error provides a measure of the average distance of each data point from the trendline. Here, we use the standard error to indicate the range of uncertainty in the trendlines for 1980 to 2020 and 1995 to 2015, and we provide p values for the linear regression analysis used to generate the trendlines. Importantly, even if we use the lower bound on the trendlines plotted in Fig. 3a, we still observe a positive trend in the number of AR events over both time periods.**

---

## Author Comment (AC3)

**Climatology and Surface Impacts of Atmospheric Rivers on West Antarctica**

Michelle L. Maclennan, Jan T. M. Lenaerts, Christine A. Shields, Andrew O. Hoffman, Nander Wever, Megan Thompson-Munson, Andrew C. Winters, Erin C. Pettit, Theodore A. Scambos, and Jonathan D. Wille

**REVIEWER COMMENT #3:**

General Comments

This is a really interesting paper that investigates the role of atmospheric rivers on the surface mass balance of the West Antarctic ice sheet, for which there is a clear knowledge gap. The authors have obviously put in a lot of work into this and strived for high standards. The Introduction is nicely written/researched, and the study nicely put into context with previous work. However, Section 2 on data and methods is difficult to follow as it is disorganised / disjointed and contains sometimes unnecessary text – the paper would really be improved if this section could be organised better. Section 3 is a well written description of the comprehensive analysis. The figures are clear and appropriate. I was slightly uncertain about Figs 2 and 3, as the results mentioned in the text did not seem to be in the same range as Fig. 2, and the justification for the trend 1995-2015 was not clear in Fig. 3 – also a possible explanation for these trends seems to be missing. But despite that the authors have obviously put a lot of work into this analysis. The study includes a very comprehensive, well researched, and well considered Discussion section which does a very good job of contextualising the results. To summarise, I think this is an excellent study, but would benefit from addressing some of the comments below, especially related to Section 2 which really needs to be much clearer / linear – especially given the complexity of the analysis and the number of data sets and the incorporation of both climatological and case study analysis.

**We thank the reviewer for taking the time to review this manuscript and providing recommendations to improve the structure and organization of the text. In response to the reviewer's recommendations below, we have revised the methods section to stick with describing each dataset in its respective subsection. Furthermore, we provide explanations and context to clarify the information presented in Figures 2 and 3. Responses are written in bold, and excerpts from the manuscript are *italicized*. Changes to the text are *italicized and in blue*. Line numbers refer to track changes in the revised manuscript. Our responses are as follows:**

Specific Comments

+ The motivation for the case study in the Introduction is not that clear. I understand that it is included as it can be investigated in more detail using the in-situ observations, and so

complements the more broader scale climatology work.  But this is not that well explained and comes across as rather disjointed.  Please strengthen this justification.

> **Reviewer Comment #1 mentioned a similar need to improve motivation and clarity in the introduction, and we have revised the last two paragraphs as a result. The final paragraph of the introduction (starting on line 70) explains the motivation for our climatology and case study and highlights the gap in prior research that our study addresses. We explain that based on previous studies, the spatial variability of extreme precipitation associated with ARs over West Antarctica is poorly understood and fails to capture the local accumulation associated with AR events. We highlight that our analysis of a case study event provides key indications of small-scale spatial variability in AR-driven accumulation and surface melting on Thwaites Eastern Ice Shelf. Finally, we emphasize that placing this case study within the broader context of the climatology of West Antarctic ARs enables us to better understand the characteristics and impacts of ARs on the surface mass balance.**

+ Section 2.1 is labelled 'observations' but has quite a few sentences describing the method, including the SNOWPACK model which is mentioned before described in its own dedicated subsection later on.  I find this rather unstructured/confusing/disorganised and would suggest that a dedicated methodology section would help the reader.   And in general, please choose appropriate sub-headings and stick to the appropriate content for these headings.

> **We have removed references to the atmospheric reanalyses MERRA-2 and ERA5 and SNOWPACK modeling in section 2.1 to improve clarity and reduce confusion. We now stick to discussing the available observations and how we use them for the AR case study event in February 2020. In general, we have revised the Data and Methods (section 2) to ensure the content of each of the four subsections is consistent with their respective sub-headers.**

+ Lines #98 - #100: More details on the reanalysis are required such as their appropriateness / representativeness of the AIS, and even just spatial resolution are necessary. Also, the reanalysis are compared with the in-situ observations on Thwaites, but there is no explanation for whether this is appropriate. For example, whether the in-situ observations are representative of a wider area that is comparative to the reanalysis grid boxes.

> **We have moved the introduction of the reanalysis data from section 2.1 to section 2.2, titled "Reanalysis Products: MERRA-2 and ERA5". We have re-organized section 2.2 to improve the flow and clarity of this section. First, we introduce both datasets. Then we describe how we use the data in order of**

**how the results are presented (first the West Antarctic AR climatology, and then the February 2020 case study).**

**Following the introduction of each reanalysis product and their spatiotemporal coverages, we added a sentence to justify our use of MERRA-2 to analyze the drivers and impacts of West Antarctic AR events (line 114):**

*We primarily use MERRA-2 analyze the large-scale synoptics and impacts of AR events in West Antarctica, as MERRA-2 explicitly represents ice sheet hydrological and energy budgets and compares best to ice core records of snow accumulation in Antarctica among multiple reanalyses (Gelaro et al., 2017; Medley and Thomas, 2019).*

**In the last paragraph of section 2.2, which discusses the use of MERRA-2 and ERA5 in the case study event, we have added the following (line 125):**

*While AMIGOS observations reflect local conditions at the Cavity and Channel Camp sites, MERRA-2 and ERA5 data represent grid-cell averages, meaning local values for temperature, surface pressure, wind speed and wind direction can deviate from those grid-cell averages. In the near-surface temperature comparison, MERRA-2 and ERA5 use 2 m temperatures while the observed temperatures are from approximately 6 m above the surface. We include ERA5 in this analysis because there are differences between MERRA-2 and ERA5 in 2 m temperature and snow accumulation during the event.*

**The spatial resolution of each reanalysis is mentioned in the first paragraph of section 2.2.**

**Finally, we have revised the introduction to include that comparing atmospheric reanalyses to the observations during the February 2020 event is a goal of this study - we want to know how well reanalyses are able to reproduce this event (line 77):**

*Then, we use in-situ observations and a firn model to examine the specific impacts of a series of three successive ARs that made landfall on TG in February 2020, as well as the ability of reanalyses to reproduce those observations.*

+ Section 2.2: Please see comment above about discussing reanalysis data before it is properly introduced. Another comment here is that you state that the datasets are 'regularly gridded', so is that in terms of lat/lon? Also, much of the text in this section again seems rather inappropriate and better placed elsewhere. For example, mention that 'this region' has experienced large acceleration in recent years should surely have been clarified in the Introduction and no need for repetition. Finally, its not really clear why MERRA is used for

one purpose (as opposed to ERA5) and ERA5 only used for comparison with MERRA during the case study.

> **"Regularly-gridded" refers to the spatial coverage of the data, which are on a latitude-longitude grid.**
>
> **We have removed the text about the accelerating mass loss from West Antarctica, which was already discussed in the introduction.**
>
> **We have added several sentences throughout section 2.2 explaining why and when we use MERRA-2 reanalysis vs ERA5; please see the previous comment for details.**

+ Section 2.4: This section is labelled SNOWPACK firn modelling but the opening sentence discusses precipitation from reanalysis. Please restructure these sections much better.

> **We have removed references to MERRA-2 and ERA5 reanalyses from the opening sentence of section 2.4 (line):**
>
> *We use observed snow height and temperature from the AMIGOS to force the firn model SNOWPACK (Lehning et al., 2002a, b) to reconstruct accumulation and surface melt during the AR case study event in February 2020.*
>
> **Overall, we have revised the section to ensure we focus specifically on SNOWPACK firn modeling and how we use it in the study.**

+ Section 2.5: Its not clear why surface height changes using interferometric reflectometry is necessary given that the in-situ observations also mention snow height. Can you please clarify?

> **As shown in Fig. 1, the GNSS-interferometric reflectometry sites are located inland from Thwaites Eastern Ice Shelf, at Lower and Upper Thwaites Glacier. The records from these sites provide more information on spatial variability and local accumulation over Thwaites Glacier, not only on the ice shelf. To clarify, we have revised the text in section 2.5 on lines 175 and 182:**
>
> *We supplement the record of surface height change estimates observed by the AMIGOS on Thwaites Eastern Ice Shelf with surface height change measurements from the grounded TG, observed with the global navigation satellite system (GNSS) using interferometric reflectometry (Larson et al., 2009, 2015; Roesler and Larson, 2018).*
>
> *The addition of GNSS-IR snow accumulation records enables us to compare spatial differences in snowfall on Thwaites Eastern Ice Shelf and TG during the AR case study event.*

+ Section 3.1, first paragraph: 1) The value given is 3.2% but Figure 2 only shows AP frequency values from 0 to 0.8%? So its not at all clear how this value was calculated. 2) Please clarify how the uncertainty value is computed? 3) Similar to the above, its not clear where the value of 28.7% comes from as this is not the range in Figure 2.

**We have added a sentence to the beginning of the paragraph to introduce AR frequency and how it is calculated, and reference Fig. 2 later in the paragraph when we discuss local AR frequency (line 195). 3.2% refers to the total AR frequency over the whole region (i.e., there is an AR making landfall somewhere in the region 3.2% of the time, on average). 0.2-0.8% refers to local AR frequency at a given point within the region, with 0.2% being the lowest value and 0.8% being the highest in the region. The uncertainty values refer to one standard deviation from the mean, which we have added into the text the first time it is used. We use r-squared from a linear regression to compute the 28.7% percent interannual variability in the total precipitation explained by AR precipitation.**

*To determine the frequency of ARs over the Amundsen Sea Embayment and Marie Byrd Land region, we divide the number of AR times by the total time from 1980 to 2020. Our analyses show that ARs exhibit a total frequency of 3.2% over the whole region from 1980 to 2020 (i.e., there is an AR making landfall somewhere in the region 3.2% of the time, on average) (Fig. 2). This represents the total frequency of ARs over the region, calculated by dividing the number of AR times by the total time from 1980 to 2020. Within the region, localized AR frequencies range from 0.2 to 0.8% of the time, with the highest frequencies over the Abbot Ice Shelf and the Getz Ice Shelf (Fig. 2a). Integrated over the entire region, ARs contribute 59 +/- (one standard deviation) 24 Gt precipitation annually (out of 550 +/- 63 Gt total annual precipitation, Fig. 2b and c), and explain 28.7% of the interannual variability in precipitation (linear trends removed).*

+ Figure 3: Is the large variability of AP events connected to the large variability in the Amundsen Sea Low / large interannual variability in cyclone frequency in this region (Simmonds and Keay, 2000)?

**While the semi-annual zonal migration of the Amundsen Sea Low drives strong seasonal variability in the total amount of snowfall on Thwaites Glacier (Maclennan and Lenaerts, 2021), we state in our results (line 211) that ARs in this region do not exhibit statistically significant seasonality in their number nor in their duration. The interannual and multi-decadal variability in the number of AR events may be explained by variability in the strength and positioning of the Amundsen Sea Low and multiple modes of atmospheric variability.**

**Maclennan, M. L. and Lenaerts, J. T. M. (2021): Large-Scale Atmospheric Drivers of Snowfall over Thwaites Glacier, Antarctica. Geophysical Research Letters, doi: 10.1029/2021GL093644**

+ Figure 3: 1) Can you please justify why the range 1995 to 2015 was chosen?  Bluntly, was this cherry picked to get a significant correlation? What if you shifted the range by 1 or 2 years, how does the trend change and its significance? 2) There doesn't seem to be any mention of what could be causing the positive trend in AR events – this is also noticeably absent from the Discussion. For example, could this be due to decadal changes in the Madden-Julian Oscillation (Hsu et al., 2021; Science Advances) which occurred in the late twentieth century and early twenty-first century?

1) **Yes, we can justify the range of 1995 to 2015 which was chosen to highlight multi-decadal variability in the number of AR events within the longer-term positive trend from 1980 to 2020. The long-term trend from 1980 to 2020 is 0.12 +/- 0.06 events per year squared (p = 0.055, standard error = 0.0595). This represents a statistically significant increase in the annual number of AR events over time. Within the 1980 to 2020 period, however, there is shorter-term variability in the number of AR events over time as well. We selected the 1995 to 2015 period to highlight this short-term variability because it exhibits a statistically significant positive trend that is notably higher than the total trend from 1980 to 2020. From 1995 to 2015, the trend in AR events is 0.32 +/- 0.16 events per year squared (p = 0.059, standard error = 0.1598).**

   **We performed the same statistical analysis on 20-year periods within the five years before and after 1995 to 2015 (i.e., 1990-2010, 1991-2011, …, 1999-2019, 2000-2020). Among the periods tested, 1995-2015 exhibits the lowest standard error, a low p value, and a high trend combined. 1996-2016 similarly exhibits a high trend and low p value, but with a slightly higher standard error (still statistically significant). That is why we selected the 1995 to 2015 range. We have revised the text as follows to highlight the role of shorter-term variability within the overall trend (line 203):**

   ***From 1980 to 2020, there is a positive trend in AR events of +0.12 +/- 0.06 events per year squared (p = 0.055)****, similar to the results from Wille et al. (2021), which also showed an increasing trend in AR frequency from 1980 to 2018 over the WAIS region****. From 1995 to 2015, there is a marked trend of +0.32+/- 0.16 events per year squared (p = 0.059)****, indicating multi-decadal variability in the number of AR events embedded within the longer-term positive trend (this 20-year period is selected based on its high trend, low p value, and low standard error of 0.16 events per year squared)****.***

2) **There are a number of modes of variability, both decadal and interannual, that impact this region, most notably the PSA2 - for references on phase sign for each year in the record, see supplemental figure S2 in Shields et al. (2022). However, the rarity of AR events, combined with the interaction of multiple modes of variability, makes it challenging to link specific trends in AR activity to individual modes (Wille et al., 2021; Shields et al., 2022). We have revised the Discussion to include this point and highlight that future research on how modes of variability and anthropogenic forcing will be critical to understanding how ARs and their impacts may change in the future (line 410):**

*While AR events occur slightly more frequently over the Antarctic Peninsula and Dronning Maud Land than over the Amundsen Sea Embayment and Marie Byrd Land (Wille et al., 2021), the vulnerability of the latter region to ocean-induced ice mass loss and ice sheet instability amplifies the importance of quantifying accumulation and the interannual variability of AR events, as well as the modes of atmospheric variability driving their long-term trends (Shields et al., 2022), as a compensation mechanism for the mass loss. The long-term positive trend in the number of AR events and the shorter-term variability identified in this study underlines the importance of understanding how modes of atmospheric variability, especially the PSA2, and anthropogenic forcing are impacting AR activity in this region (Dalaiden et al., 2022).*

**Shields, C., Wille, J., Collow, A., Maclennan, M., and Gorodetskaya, I. (2022): Evaluating Uncertainty and Modes of Variability for Antarctic Atmospheric Rivers. Geophysical Research Letters., doi: 10.1029/2022GRL09957**

+ Could the pressure patterns / anomalies responsible for Ars be compared to the analysis of Scott et al. (2019; Journal of Climate) , which uses ERA5 and a cluster technique to identify dominant circulation patterns.  Perhaps this would be appropriate for the Discussion section.

**Scott et al. (2019) identifies the PSA2 signature consisting of a high-low pressure couplet off the coast of West Antarctica, with anomalously high 2 m temperatures pushing towards the Amundsen Sea Embayment from the Southern Ocean. Maclennan et al. (2021) found that snowfall events on Thwaites Glacier are moderately correlated with the PSA2 pattern as well. We have added a sentence on this topic to the discussion (line 324):**

*We find that AR events making landfall in the Amundsen Sea Embayment and Marie Byrd Land are driven by the coupling of a blocking high over the Antarctic Peninsula with a low-pressure system known as the Amundsen Sea Low. This pressure pattern is similar to the Pacific South-American patterns identified by (Scott et al., 2019) as drivers of marine air intrusions and West*

*Antarctic surface melting, and consistent with geopotential height anomalies identified by (Adusumilli et al., 2021) during WAIS AR events in 2019.*

Minor / Technical Corrections

+ Line #9: 3 -> three

**Done.**

***Next, we use observations from automatic weather stations on Thwaites Eastern Ice Shelf with the firn model SNOWPACK and interferometric reflectometry to examine a case study of three ARs that made landfall in rapid succession from February 2 to 8, 2020, known as an AR family event.***

+ Line #9: Please give the year of the case study.

**Done - please see previous comment.**

+ Line #13: I assume the accumulation value is water equivalent. Maybe state this.

**We have added "or millimeters water equivalent" after the units.**

+ Line #26: As written this states that all mass loss is from the WAIS, which is not the case as the Peninsula region has surely also lost mass.

**We have revised the sentence to suggest that most, but not all, mass loss is from the WAIS (line 18):**

***In the last four decades, the AIS has experienced increased mass loss, from 40 +/- 9 Gigatons per year (Gt yr$^{-1}$) between  1979 and 1990 to 252 +/- 26 Gt yr$^{-1}$ between 2009 and 2017,  most of which is attributed to increasing discharge across the grounding line of the West Antarctic Ice Sheet (WAIS, Rignot et al., 2019).***

+ Line #28: This statement requires a reference.

**We have added a reference (line 22):**

***Although it covers only 17% of the AIS, the WAIS accounts for 34% of ice discharge (Rignot et al., 2019).***

+ Line #33: TG is undefined.

**We have revised the sentence to define TG as Thwaites Glacier (line 27):**

*In particular, Thwaites Glacier (TG), which borders the Amundsen Sea, is at considerable risk for continued grounding line retreat in the future because it is grounded on inward sloping bedrock, which may lead to a rapid positive feedback for increasing ice flow and retreat, termed 'marine ice sheet instability' (Weertman, 1974; Schoof, 2012).*

+ Line #40: What about evaporation? With increasing surface melting this will become increasingly important. For example, Bromwich et al. 2011 J. Climate showed that sublimation and evaporation combined accounted for around 25% of the precipitation term.

**Here we discuss the contributors to surface mass balance from 1979 to present day. During this period, evaporation is less important than sublimation on the Antarctic Ice Sheet because there are only a few small regions where standing water is present, including some ice shelves (Kingslake et al., 2017; Langley et al., 2016; Lenaerts, Lhermitte et al., 2017). Melt water produced on snow surfaces can propagate into the firn, which prevents a significant amount of evaporation from occurring (Lenaerts et al., 2019). Bromwich et al. (2011) combines the evaporation and sublimation terms. We have changed "sublimation" to "sublimation/evaporation" to include the evaporation term in our description of surface mass balance (line 33).**

*The SMB represents the balance between mass gained at the surface through precipitation, and mass lost by sublimation/evaporation and surface meltwater runoff (Lenaerts et al., 2019).*

**Bromwich, D., Nicolas, J., and Monaghan, A. (2011): An Assessment of Precipitation Changes over Antarctica and the Southern Ocean since 1989 in Contemporary Global Reanalyses. Journal of Climate, doi: 10.1175/2011JCLI4074.1**

**Kingslake, J., Ely, J., Das, I., and Bell, R. (2017): Widespread movement of meltwater onto and across Antarctic ice shelves. Nature, doi: 10.1038/nature22049**

**Langley, E., Leeson, A., Stokes, C., and Jamieson, S. (2016): Seasonal evolution of supraglacial lakes on an East Antarctic outlet glacier. Geophysical Research Letters, doi: 10.1002/2016GL069511**

**Lenaerts, J., Lhermitte, S., Drews, R. *et al.* (2019): Meltwater produced by wind–albedo interaction stored in an East Antarctic ice shelf. Nature Climate Change, doi: 10.1038/nclimate3180**

**Lenaerts, J. T. M., Medley, B., Broeke, M. R., and Wouters, B. (2019): Observing and Modeling Ice Sheet Surface Mass Balance, Reviews of Geophysics, doi: 10.1029/2018RG000622**

+ Line #56: Mention of 'on the order of the Amazon River' is confusing. Do you mean the actual river? Is this a type of AR? Are you referring to spatial size? I'm afraid that this comparison is not that helpful so please revise.

**This is a standard analogy used to emphasize the importance of ARs in the hydrological cycle. It refers to the large quantity of water they transport, which is more than double the flow of the Amazon River. It is mentioned in Zhu and Newell (1998) and included in the American Meteorological Society's definition of an atmospheric river: [https://glossary.ametsoc.org/wiki/Atmospheric_river](https://glossary.ametsoc.org/wiki/Atmospheric_river). We have revised the text as follows (line 49):**

*ARs are associated with a low-level jet and moisture fluxes on the order of the flow of the Amazon River (Zhu and Newell, 1998).*

**Zhu, Y. and Newell, R. E. (1998): A Proposed Algorithm for Moisture Fluxes from Atmospheric Rivers. American Meteorological Society Monthly Weather Review, doi: 10.1175/1520-0493(1998)126<0725:APAFMF>2.0.CO;2**

+ Line #78: Maybe clarify this sentence a little regarding 'rely on reanalysis'. For example, by saying 'In this study, we rely ....'

**In response to Reviewer Comment #4, we have rewritten this section of the paper, including the rephrasing of "rely on reanalysis" (line 76):**

*First, we use atmospheric reanalyses to quantify the landfalls and accumulation impacts of ARs from 1980 to 2020 over Marie Byrd Land and the Amundsen Sea sector.*

+ Line #103: Its not clear whether by observations you are referring to the in situ observations or the reanalysis. See specific comment above. Please clarify your methodology/approach in a dedicated section.

**In response to comments above, we have moved all references to SNOWPACK methodology to the appropriate section (2.4). The sentence now reads as follows (line 151):**

*We use observed snow height and temperature from the AMIGOS to force the firn model SNOWPACK (Lehning et al., 2002a, b) to reconstruct accumulation and surface melt during the AR case study event in February 2020.*

+ Line #146: Is there justification for the 12 hour threshold?

**The 12-hour threshold is a parameter choice we made to define separate AR events. For this study, we tested different thresholds from 6 hours up to 36 hours,**

**and found a range of 20 events per year (6 hour threshold) to 14 events per year (36 hour threshold) on average. We found that choosing a 12-hour break period enabled us to capture the case study AR family event as comprising of three AR events, which is consistent with the poleward movement and positioning of the ARs detected during that time by the Wille et al. (2021) algorithm. Using a window shorter than 12 hours can group ARs together that are part of the same synoptic system and are not necessarily unique. Using a window longer than 12 hours risks erroneously combining multiple, unique AR events, such as those shown in the case study. An additional constraint was the 3-hourly temporal resolution of our AR detection algorithm. There are many different ways of defining ARs and AR events (Shields et al., 2018) and some studies count events using a duration, rather than time break, threshold (Fish et al., 2021). However, given the large variability in the duration of Antarctic AR events in this region, which ranges from 3 hours to days, we decided a time break was the most appropriate method for counting AR events.**

**Fish, M. A., Wilson, A. M., and Ralph, F. M. (2019): Atmospheric River Families: Definition and Associated Synoptic Conditions. Journal of Hydrometeorology, doi: 10.1175/JHM-D-18-0217.1**

**Shields, C., et al. (2018): Atmospheric River Tracking Method Intercomparison Project (ARTMIP): project goals and experimental design. Geosci. Model Dev., doi: 10.5194/gmd-11-2455-2018**

+ Line #206: Its not clear how these average surface pressure maps during AR events are computed. See comments above. Presumably you identified the ARs and then did calculated a composite of these events. But this really needs to be made clearer.

**Line 117 of section 2.2 in the Data and Methods states how the surface pressure composite maps are generated:**

***We use MERRA-2 reanalysis to generate surface pressure and surface pressure anomaly (relative to 1980 to 2020 climatology) composite maps during the times of AR landfalls over coastal West Antarctica, including the Amundsen Sea Embayment and Marie Byrd Land.***

+ Figure 4: The stippling wasn't really obvious. Could this be made clearer?

**Yes - we have increased the size of the stippling to make it clearer:**

[Figure]

+ Line #219: 1) So you are creating a distribution of the temperatures. Perhaps this needs a little more explanation. 2) What does 'all seasons' mean? Figure 5 only shows the seasonal breakdown?

**On line 224, we state how the temperature difference is calculated:**

*To further examine the impacts of AR landfalls on TG surface conditions, we calculate the change in surface temperatures on Thwaites Eastern Ice Shelf during AR events (Fig. 5). To do this, we take the difference between the mean MERRA-2 2 m temperature 24 hrs before landfall, and the mean 2 m temperature 24 hrs after landfall.*

**The distributions in Fig. 5 indicate the range of temperature differences we calculate for AR events, divided by season. We explain the results as follows. First, we discuss the overall temperature difference distribution of ARs among all seasons (not divvying up the ARs by season yet, line 226):**

*AR events are associated with a temperature increase of 1.4 K (first quartile) to 7.1 K (third quartile), with median 3.8 K, over Thwaites Eastern Ice Shelf over all seasons.*

**Then, in the following sentence, we look at the seasonal breakdown, which is presented in Fig. 5:**

*In austral summer (December-January-February), the median temperature increase is the smallest at 1.5 K. In fall (March-April-May), winter (June-July-August), and spring (September-October-November), the median temperature increases associated with AR landfall are 4.3 K, 6.3 K, and 4.3 K, respectively.*

+ Line #225: Maybe state melting point of snow/ice.

**The melting point is stated in the next sentence of the paper (line 233):**

*There are many more summer events where 2 m temperatures exceed the melting point of 273.15 K (6 events in 1980-2020) than in fall (2 events), winter (1 event), and spring (0 events).*

+ Line #291: 2 -> two

**We have changed "2" to "two" (line 299):**

*Overall, surface melt is nearly two orders of magnitude lower than the snowfall, indicating that the primary impact of this AR family event is to contribute snowfall to TG.*

+ Line #317: Again, what is the uncertainty mentioned here. Is it one standard deviation? Please clarify.

**Yes, "17 +/- 5 AR events per year" refers to one standard deviation. Please see response to previous comment on the uncertainty - we now introduce the uncertainty as one standard deviation the first time it is used.**

---

## Author Comment (AC4)

**Climatology and Surface Impacts of Atmospheric Rivers on West Antarctica**

Michelle L. Maclennan, Jan T. M. Lenaerts, Christine A. Shields, Andrew O. Hoffman, Nander Wever, Megan Thompson-Munson, Andrew C. Winters, Erin C. Pettit, Theodore A. Scambos, and Jonathan D. Wille

**REVIEWER COMMENT #4:**

General comments

The manuscript by Maclennan et al. presents a study of atmospheric river events and their effects on the climatology and surface mass balance in West Antarctica. First, the MERRA-2 and ERA5 reanalysis products are used to quantify the frequency, trends, and large-scale effects of ARs on precipitation in the period 1980 to 2020. Then in-situ observations from weather stations at Thwaites Glacier are used to reconstruct accumulation and firn conditions during a series of AR events in 2020. Finally, the possible future effects of increasing AR intensity and frequency on surface conditions and surface mass balance in the areas are discussed.

The paper provides a good background of ARs in West Antarctica and their effects on surface mass balance, and the large-scale study is combined with the in-situ data into a very interesting discussion. The topic is timely and the paper is suitable for The Cryosphere.

The only minor issue is that the presentation of the work should more clearly state the goal of the investigations as well as summarize the findings more clearly. As I see it, the main strength and new contribution of this paper is that the authors combine the large-scale reanalysis products with detailed in-situ data. Thereby, they are able to qualify the discussion of the future impacts much more convincingly than from reanalysis products alone. This message should be communicated more clearly. The discussion section is strong, but I suggest that the Discussion and Conclusion section is divided into two, so there is a separate conclusion section in order to communicate the findings more clearly.

> **The authors would like to thank the reviewer for providing comments and feedback that help to improve the content and clarity of the manuscript. In response to comments about the goal of the investigations in the introduction (which are similar to feedback provided by the other two reviewers), we have revised the last two paragraphs of the introduction to place our work in the context of previous Antarctic AR studies, highlight the gap in the existing research, and note how our study provides a key link between large-scale AR patterns over West Antarctica and localized impacts over Thwaites Glacier. Similarly, we have revised the first paragraph of the discussions and conclusion to emphasize that the combination of observation and reanalyses enables us to discuss how AR impacts may become exacerbated in a future climate. We have decided to keep the discussion and conclusions section**

**combined, because it enables us to integrate the most important findings of this study with a discussion on how our results relate to previous studies on Antarctic ARs and how they depend on our choice of methodology. Responses are written in bold, and excerpts from the manuscript are** *italicized.* **Changes to the text are** *italicized and in blue.* **Line numbers refer to track changes in the revised manuscript. Please see our responses to the comments below:**

Detailed comments:

Page 1: the abstract is far too long. The length should be 250 words (see instruction in the TC). Remove sentences that are essential background or discussion.

**The authors concur with this comment and we have reduced the length of the abstract to 250 words. Please see the revised abstract in lines 1-14 of the manuscript.**

Page 2-3: The introduction contains the motivation and background on ARs. However, the purpose of the study is not clearly stated. Rewrite the last paragraph to start with "In this study, we… This would also make it clear from the start how this paper differs from earlier studies by including the in-situ data, and why these data are included.

**We have rewritten the last paragraph of the introduction (line 70) to clarify the purpose of our study and how it addresses a knowledge gap in prior studies. The last paragraph has been rewritten as follows:**

*Despite the importance of ARs for WAIS SMB, the spatial variability of extreme precipitation associated with ARs over Antarctica is poorly understood. Previous research using a regional climate model showed coastal regions of the WAIS broadly experience 1-3 days of AR conditions per year which account for around 40% of extreme precipitation events from 1980 to 2020 (Wille et al., 2021). However, the low spatiotemporal resolution of atmospheric reanalysis products does not highlight the effects of topography or capture precipitation patterns during individual AR events (Gehring et al., 2022). In this study, we provide both a large-scale, climatological perspective of West Antarctic ARs and a focused case study of a particular AR event. First, we use atmospheric reanalyses to quantify the landfalls and accumulation impacts of ARs from 1980 to 2020 over Marie Byrd Land and the Amundsen Sea sector. Then, we use in-situ observations and a firn model to examine the specific impacts of a series of three successive ARs that made landfall on TG in February 2020, as well as the ability of reanalyses to reproduce those observations. Our analysis provides key indications of small-scale spatial variability in AR-driven accumulation and surface melting on TG, within the broader context of the climatology of ARs in the region. Finally, we discuss*

*how ARs contribute to the present mass balance of the WAIS, which improves our understanding of how their impacts may change in future climate scenarios.*

Page 2:, line 33: I don't think "TG" has been defined, please do so.

**Thank you for pointing this out, we have revised the sentence to define TG as Thwaites Glacier (line 27):**

*In particular, Thwaites Glacier (TG), which borders the Amundsen Sea, is at considerable risk for continued grounding line retreat in the future because it is grounded on inward sloping bedrock, which may lead to a rapid positive feedback for increasing ice flow and retreat, termed 'marine ice sheet instability' (Weertman, 1974; Schoof, 2012).*

Page 4: AMIGOS – include a reference to define what AMIGOS is, it is not enough to include it in the title of section 2.1

**We have removed AMIGOS from the title of section 2.1 and renamed it "Observations from Automatic Weather Stations". In the first sentence of section 2.1, we introduce the AMIGOS as follows (line 85):**

*Through the Thwaites-Amundsen Regional Survey and Network Integrating Atmosphere-Ice-Ocean Processes (TARSAN) project of the International Thwaites Glacier Collaboration, automatic weather stations known as Automated Meteorology–Ice–Geophysics Observation System (AMIGOS, Scambos et al., 2013) were installed on Thwaites Eastern Ice Shelf at Cavity Camp (75.033 °S, 105.617 °W) and Channel Camp (75.050 °S, 105.4334 °W) during a field campaign in austral summer 2019/20 (Fig. 1).*

Page 4, line 100: add "s" to sensor, and change "is" to "was".

**We have revised the sentence to reflect that there were two sensors (plural) at 6 m above the surface during the event (line 95).**

*The AMIGOS temperature sensors were located about 6 m above the surface during the period of interest in this study, so we refer to AMIGOS air temperatures as "near-surface" when compared to 2 m air temperatures from MERRA-2 and ERA5.*

Page 5: Perhaps explain a little more clearly why you focus on the precipitation and use the vIVT algorithm to detect the ARs. The precipitation effect is most important at present, but this could perhaps be made more clear here, or stated earlier in the motivation.

We have added clarification in section 2.3 to emphasize why we use the vIVT algorithm to detect ARs (starting on line 137):

*Wille et al. (2021) found that IWV is better suited for identifying ARs that cause surface melting, as high IWV over the AIS is associated with cloud development and high downwelling longwave radiation to the surface. Comparatively, the vIVT-based definition of ARs is better suited for studying snowfall, since the meridional transport of water vapor is linked to atmospheric dynamics that lead to precipitation.*

*In this study, we primarily focus on AR-driven precipitation, and thus we use the vIVT catalogues with AR detection at 3 hourly intervals based on MERRA-2 reanalysis.*

We have also revised the last two paragraphs of the introduction to clarify the motivation for studying AR-driven precipitation, starting on line 58.

Page 7, figure 1: Indicate the 80degS latitude at the figure to the left. This would be helpful later in the discussion. What is the black outline in the middle figure?

We have added information on the 80 deg S boundary for AR detection to Fig. 2 in the paper, which shows results from the Wille et al. (2021) AR detection algorithm and marks the 80 deg S boundary with a grey mask. The figure caption for Fig. 2 has been revised as follows:

*Grey shading over the interior of the ice sheet marks the 80 deg S boundary of the Wille et al. (2021) AR detection algorithm.*

The black outline in the middle figure of Fig. 1 refers to the region of interest in this study. We have added it to the first sentence of the figure caption to improve clarity:

*Map showing the region of interest in this study -- the Amundsen Sea Embayment and Marie Byrd Land in West Antarctica (outlined in black).*

Page 10, line 231: Please define "ASE".

We have removed ASE (Amundsen Sea Embayment) from the sentence and replaced it with "Amundsen Sea sector" (line 238).

*The geometry and orientation of TG render it highly susceptible to synoptic flow-induced snow storms caused by ocean air masses that are driven from the Southern Ocean into the  Amundsen Sea sector.*

Page 12, line 278: remove "and".

**Done (line 283).**

***MERRA-2 accumulation is 88 kg m$^{-2}$,  ERA5 accumulation is 87 kg m$^{-2}$.***

Page 12, line 280: The spatial resolution of the reanalysis product could both mean that it does not resolve variations within the grid cell, and also that some larger scale patterns are not resolved properly. It could be relevant to mention both.

**Sub-grid scale processes, including local snowfall over Thwaites Eastern Ice Shelf, may not be resolved by the reanalysis products used in this study. However, we are confident that the reanalysis is able to capture the synoptic-scale features of interest that we discuss in the paper. Previous studies including Turner et al. (2013) and Raphael et al. (2016) have used reanalyses to describe large-scale flow patterns in this region. Gorodetskaya et al. (2014), Pohl et al. (2021), and Turner et al. (2022) used reanalyses to examine the synoptic forcing of atmospheric rivers in other regions of Antarctica.**

**In section 2.2 we have added discussion on how the difference in spatial resolution between the reanalyses and observations means that the reanalyses may not resolve local weather at the AMIGOS sites (line 125):**

*While AMIGOS observations reflect local conditions at the Cavity and Channel Camp sites, MERRA-2 and ERA5 data represent grid-cell averages, meaning they may under- or over-estimate local values for temperature, surface pressure, wind speed and wind direction slightly when compared to the observations.*

**Gorodetskaya, I. V., Tsukernik, M., Claes, K., Ralph, M. F., Neff, W. D., and Van Lipzig, N. P. M. (2014): The role of atmospheric rivers in anomalous snow accumulation in East Antarctica, Geophysical Research Letters, doi: 10.1002/2014GL060881**

**Pohl, B., Favier, V., Wille, J., Udy, D. G., Vance, T. R., Pergaud, J., Dutrievoz, N., Blanchet, J., Kittel, C., Amory, C., Krinner, G., and Codron, F. (2021): Relationship Between Weather Regimes and Atmospheric Rivers in East Antarctica. Journal of Geophysical Research: Atmospheres, doi: 10.1029/2021JD035294**

**Raphael, M. N., Marshall, G. J., Turner, J., Fogt, R. L., Schneider, D., Dixon, D. A., Hosking, J. S., Jones, J. M., and Hobbs, W. R. (2016): The Amundsen Sea Low: Variability, Change, and Impact on Antarctic Climate. Bulletin of the American Meteorological Society, doi: 10.1175/BAMS-D-14-00018.1**

Turner, J., Phillips, T., Hosking, J. S., Marshall, G. J., and Orr, A. (2013): The Amundsen Sea low. International Journal of Climatology, doi: 10.1002/joc.3558

Turner, J., Lu, H., King, J. C., Carpentier, S., Lazzara, M., Phillips, T., and Wille, J. (2022): An Extreme High Temperature Event in Coastal East Antarctica Associated With an Atmospheric River and Record Summer Downslope Winds. Geophysical Research Letters, doi: 10.1029/2021GL097108

Page 14, line 306: please correct the reference.

**The reference has been corrected (line 316):**

***Using the Wille et al., (2021) AR detection algorithm based on vIVT, we are able to diagnose the local climatology of ARs making landfall over the Amundsen Sea Embayment and Marie Byrd Land.***

Page 14, line 312: Add the 80degS latitude to figure 1, see comment above.

**Please see response to previous comment, we have added a description of the 80 deg S boundary to Fig. 2, where it is shown through a grey mask over the interior of the Antarctic Ice Sheet.**

Page 17: I miss a conclusion section to summarize the findings clearly and provide an outlook.

**Section 4 combines the discussion and conclusions from the study. Based on comments from all reviewers that this section is interesting and compelling, we as co-authors have considered this option and agreed that the discussion and conclusions section is best left as it is currently formatted. By setting our conclusions in the context of broader Antarctic AR and surface mass balance research, we show how this study is part of a larger research aim to identify Antarctic ARs, diagnose their synoptic characteristics, and quantify their impacts on precipitation and surface melt. From the choice of AR detection algorithm and reanalysis products, to the AR climatology we derive, we strive to explain what critical choices were made in the data and methods, and how those may impact the results presented. Similarly with the case study, we seek to summarize how the collection of observations and reanalyses help to provide a more complete picture of the event than any single dataset alone. By combining the conclusions with the discussion, we integrate the most important findings from the study with the strengths and limitations of our methodology.**

In the final two paragraphs of the section, we provide an outlook for future research on the impacts of ARs on West Antarctic surface mass balance. We discuss the dominance of snowfall over surface melting in current ARs and how more extensive rainfall and surface melting may occur in a future climate, particularly if AR intensities are amplified. We discuss the potential implications of rainfall and melting to reduce the ability of the firn layer to absorb meltwater, which may lead to the destabilization of ice shelves and accelerated mass loss. We consider that future impacts of Antarctic AR may approach the present-day impacts of Greenland ARs. Finally, we highlight the critical need to examine the representation of Antarctic ARs in climate models and how their frequency, intensity, and surface mass balance impacts may change in the future.